# Combining $CO_2$ reduction with propane oxidative dehydrogenation over bimetallic catalysts

Elaine Gomez [1], Shyam Kattel[2], Binhang Yan[2], Siyu Yao[2], Ping Liu[2] & Jingguang G. Chen[1,2]

The inherent variability and insufficiencies in the co-production of propylene from steam crackers has raised concerns regarding the global propylene production gap and has directed industry to develop more on-purpose propylene technologies. The oxidative dehydrogenation of propane by $CO_2$ ($CO_2$-ODHP) can potentially fill this gap while consuming a greenhouse gas. Non-precious FeNi and precious NiPt catalysts supported on $CeO_2$ have been identified as promising catalysts for $CO_2$-ODHP and dry reforming, respectively, in flow reactor studies conducted at 823 K. In-situ X-ray absorption spectroscopy measurements revealed the oxidation states of metals under reaction conditions and density functional theory calculations were utilized to identify the most favorable reaction pathways over the two types of catalysts.

[1] Department of Chemical Engineering, Columbia University, New York, NY 10027, USA. [2] Chemistry Department, Brookhaven National Laboratory, Upton, NY 11973, USA. Correspondence and requests for materials should be addressed to J.G.C. (email: jgchen@columbia.edu)

Propylene is one of the most diverse petrochemical building blocks used for the production of many chemicals (e.g., polypropylene, propylene oxide, and acrylonitrile). The co-production of propylene from steam and fluidized crackers is anticipated to be insufficient to satisfy the rapidly growing demand[1]. Consequently, there is a need for the development of economic on-purpose production techniques to produce additional propylene. The direct dehydrogenation of propane (DDP) is thermodynamically limited and is highly endothermic ($\Delta H^{\circ}_r = 29.70$ kcal/mol), requiring temperatures that may exceed 973 K for significant propylene yields[2]. In principle, the introduction of $CO_2$ as a mild oxidant into the feed alters the dehydrogenation pathway by oxidizing the abstracted hydrogen from the alkane and consequently releasing the heat of reaction that reduces operating temperatures[2,3]. The presence of $CO_2$ can also increase the equilibrium conversion of propane by consuming $H_2$ through the reverse water gas shift reaction (RWGS), as seen in the thermodynamic calculations in Fig. 1a. Additionally, unlike regular oxidative dehydrogenation with molecular oxygen, $CO_2$ as a mild oxidant suppresses over-oxidation and thus minimizes the production of carbon oxides. The reactions of propane and $CO_2$ also have the potential to employ two underutilized[4–6] reactants to supply propylene as well as to mitigate detrimental $CO_2$ emissions[7,8].

The reactions of $CO_2$ with propane may occur through two distinct pathways, oxidative dehydrogenation ($CO_2 + C_3H_8 \rightarrow C_3H_6 + CO + H_2O$) and dry reforming ($3CO_2 + C_3H_8 \rightarrow 6CO + 4H_2$). The two reactions should occur simultaneously at temperatures around 823 K and above with considerable conversions (Fig. 1b), allowing the formation of both dehydrogenation products (propylene) and reforming products (synthesis gas). The oxidative dehydrogenation of propane by $CO_2$ ($CO_2$-ODHP) can reach an equilibrium conversion of 33% as opposed to 17% for DDP at 823 K. At that same reaction temperature, as seen in Fig. 1c, $CO_2$ equilibrium conversion for the dry reforming of propane (DRP) can reach up to 98% at a temperature 150 K less than that of methane dry reforming (DRM). This in turn would reduce catalyst deactivation due to coking and phase transformations triggered by the relatively high temperatures commonly used in DRM[9,10]. Furthermore, in the $CO_2+C_3H_8$ system unreacted $CO_2$ can remove surface carbon via the Boudouard reaction ($CO_2 + C_s \rightarrow 2CO$) at temperatures as low as 773 K with moderate rates[11,12]. Thus, it is of great interest to identify catalysts that can either selectively break the C–H bond to produce propylene or the C–C bonds to generate synthesis gas (CO + $H_2$).

Previous work in $CO_2$-ODH primarily focuses on supported chromium catalysts[13–15] as a result of their ability to exist in multiple oxidation states[16], but implementation is limited due to short lifecycles and high toxicity of chromium[17]. Ni is mainly used

for dry reforming, but catalyst deactivation due to severe coking is still a problem[18–20]. To alleviate coke formation, precious metal catalysts (e.g., Rh, Re, Ru) have also been investigated on high surface area $Al_2O_3$[21,22]. However, large scale catalytic conversion of $CO_2$ into valuable products would require the development of cost effective, selective, and coking-resistant catalytic systems. While there are studies that examine the $CO_2$-ODHP or DRP separately, a thorough examination utilizing supported bimetallic catalysts at a temperature range that allows both pathways to occur is still lacking. Ceria ($CeO_2$) is a good choice of oxide support because it has the ability to store/release oxygen and thus may induce direct C–O bond scission of $CO_2$, while also providing available lattice oxygen for coke suppression[9,23–25].

The present work will explore ceria supported bimetallic catalysts, non-precious metal $Fe_3Ni$ as well as precious metal-based $Fe_3Pt$ and $Ni_3Pt$, that are active at 823 K. In summary, steady-state flow reactor studies indicate that $Fe_3Ni$ shows promising selectivity toward propylene via the $CO_2$-ODHP pathway, whereas $Ni_3Pt$ is active for the DRP with high selectivity toward CO. Density functional theory calculations of the energetics for the C–H and C–C bond scissions over the two catalysts are in agreement with experimental results.

## Results

**Catalytic evaluation with kinetics and deactivation patterns.** Flow reactor studies measuring both $CO_2$-ODHP and DRP activity simultaneously are summarized in Table 1 along with CO chemisorption values. All catalysts were synthesized via incipient wetness impregnation of metals onto commercially obtained $CeO_2$ (35–45 $m^2$/g, Sigma Aldrich). For additional details see Methods section or Supplementary Methods section. Results for conversions and product selectivity following time on stream for all catalysts are shown in Supplementary Fig. 1. The monometallic $Ni_1$ catalyst exhibits 12%–87% $C_3H_6$ and reforming selectivity, respectively, with minimal cracking products ($CH_4$ and $C_2$ hydrocarbons), while the $Fe_3$ monometallic catalyst is not active for either reaction. The bimetallic system, $Fe_3Ni$, however, at steady-state demonstrates propylene production from the $CO_2$-ODHP reaction, corresponding to 58.2% $C_3H_6$ selectivity. The differences among the propylene yields on a $C_3H_8$ basis provided in Supplementary Table 1 of $Fe_3Ni$ (1.6% $C_3H_6$ yield) and the respective monometallics ($C_3H_6$ yield of 0.4% over Ni and 0.2% over Fe) indicate that there is a synergistic effect from the formation of the bimetallic $Fe_3Ni$ catalyst.

Exchanging Ni in the $Fe_3Ni$ catalyst with precious metal Pt ($Fe_3Pt$) roughly reduces the activity by half, decreases the selectivity toward $C_3H_6$ to 32%, and is unstable compared to $Fe_3Ni$ (Supplementary Fig. 2). The other precious metal bimetallic catalyst, $Ni_3Pt$, primarily performs the DRP reaction with 39%

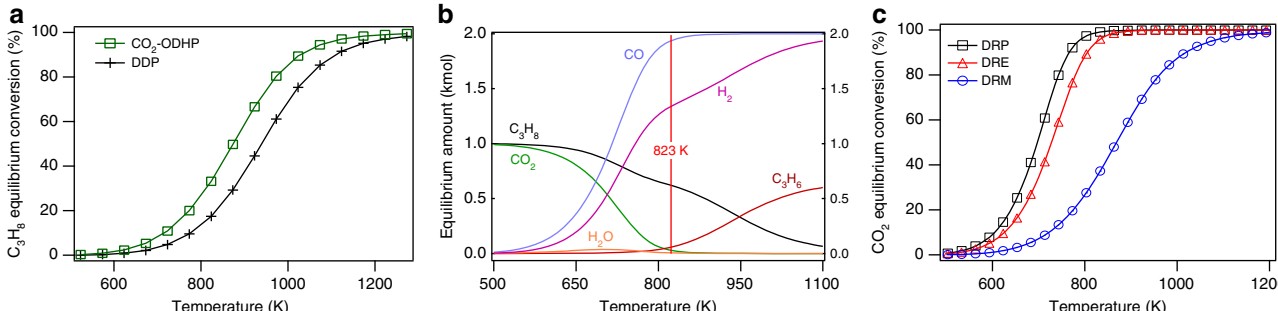

**Fig. 1** Thermodynamic equilibrium plots. Equilibrium calculations were performed through HSC Chemistry 8 software, which utilizes a Gibbs free energy minimization algorithm. **a** $C_3H_8$ equilibrium conversion for $CO_2$-ODHP and direct dehydrogenation of propane; **b** product amounts for $CO_2+C_3H_8$ system and **c** conversions of propane, ethane, and methane dry reforming; all vs. temperature at 1 atm

**Table 1 Catalyst flow reactor results for $CO_2 + C_3H_8$ reaction**

|  | Fe$_3$Ni | Fe$_3$Pt | Ni$_3$Pt | *Ni$_3$Pt | Ni$_1$ | Ni$_3$ | Pt$_1$ |
|---|---|---|---|---|---|---|---|
| CO uptake ($\mu mol\,g^{-1}$) | 31.9 | 31.5 | 50.1 | – | 13.1 | 37.7 | 16 |
| Conversion (%) |  |  |  |  |  |  |  |
| $\quad CO_2$ | 4 | 2.6 | 39.4 | 7.8 | 9.3 | 32.8 | 4.2 |
| $\quad C_3H_8$ | 2.7 | 1.1 | 11.6 | 2.2 | 3 | 9.6 | 1.6 |
| TOF (site$^{-1}$ min$^{-1}$) |  |  |  |  |  |  |  |
| $\quad CO_2$ | 5.7 | 3.5 | 37.5 | – | 31.9 | 40.2 | 8.1 |
| $\quad C_3H_8$ | 3.4 | 1.5 | 10.5 | – | 8.9 | 11.4 | 2.8 |
| Selectivity (%) |  |  |  |  |  |  |  |
| $\quad CO$ | 40.2 | 65.1 | 96.2 | 87.8 | 86.8 | 94.9 | 77 |
| $\quad C_3H_6$ | 58.2 | 32 | 2.8 | 11 | 12.3 | 2.9 | 21.2 |
| $\quad CH_4$ | 0.8 | 1.3 | 0.83 | 0.9 | 0.6 | 2.11 | 0.8 |
| $\quad C_2H_6$ | 0 | 0 | 0.1 | 0 | 0.24 | 0.05 | 0.9 |
| $\quad C_2H_4$ | 0.8 | 1.6 | 0 | 0.3 | 0 | 0.06 | 0 |
| Yield (%) |  |  |  |  |  |  |  |
| $\quad CO$ | 1.1 | 0.7 | 11.1 | 2 | 2.6 | 9.1 | 1.3 |
| $\quad C_3H_6$ | 1.6 | 0.3 | 0.3 | 0.2 | 0.4 | 0.3 | 0.4 |

10 mL/min each reactant at 823 K with Ar diluent (20 mL/min) and 100 mg of catalyst (16–20 mesh). Catalysts marked with an asterisk indicate that the sample was diluted to achieve comparable $C_3H_8$ reactant conversion to Fe$_3$Ni. Values are obtained by averaging data from 10–12 h. Selectivity and yield are on a $C_3H_8$ basis (including only carbonaceous species). Catalysts are synthesized by atomic ratios corresponding to a 1.67 wt.% Pt$_1$ basis, thus the weight percent of Fe$_3$, Ni$_1$, and Ni$_3$ are 1.43, 0.5, and 1.5, respectively. The nomenclature assigned by subscripts such as in Fe$_3$Ni means that there are three atoms of Fe for every atom of Ni

$CO_2$ conversion, a robust selectivity toward CO of 88% at comparable reactant conversions (Supplementary Table 2) and is more stable compared to monometallic Ni$_3$ (Supplementary Fig. 3). Thus, when Ni is coupled with non-precious Fe at a ratio of 1:3, higher dehydrogenation activity can be achieved and propylene is produced. In contrast, when Ni is alloyed with precious metal Pt, reforming activity is enhanced compared to monometallic Ni$_3$. Further analysis, such as the comparison of CeO$_2$ supported Ni$_3$Pt with Ni$_3$Fe and Fe$_3$Ni catalysts along with CO selectivity following $CO_2$ conversion plots can be found in Supplementary Notes 1 and 2, respectively.

Kinetic studies examining the influence of the reactant partial pressure and the reaction temperature on the activity of Fe$_3$Ni and Ni$_3$Pt were conducted to further evaluate the differences between the two types of catalysts. The apparent activation energies were derived by measuring production rates in the temperature range of 803–843 K. Over Fe$_3$Ni, the activation barrier for propane $CO_2$ oxidative dehydrogenation was found to be 115 kJ mol$^{-1}$, while the activation barrier for reforming over Ni$_3$Pt was 119 kJ mol$^{-1}$. Arrhenius-type plots and additional values are available in Supplementary Fig. 4 and Supplementary Table 3, respectively. As seen in Fig. 2a, the reactant consumption rate of $C_3H_8$ for the Fe$_3$Ni $CO_2$-ODHP catalyst was initially unaffected by increasing the partial pressure of $CO_2$ but upon reaching a $C_3H_8$:$CO_2$ ratio of 1:1, the rate started to decline. The reforming catalyst, on the other hand, was positively influenced by the partial pressure of $CO_2$ until the aforementioned ratio of 1:6. Increasing the $C_3H_8$ partial pressure produced similar trends and are shown in Supplementary Fig. 5. The declining rates signify that there are less catalytic sites available for one reactant when the other is in excess, indicative of competitive adsorption of adsorbates and/or surface intermediates. Particularly, the rates for both reactants decrease at high propane partial pressure, suggesting that as the reaction progresses intermediates from propane block surface sites and lead to a loss in activity.

To further evaluate how different reaction pathways may influence deactivation patterns, both thermogravimetric (TGA) and energy dispersive spectroscopy (EDS) experiments were conducted and results are provided in Supplementary Figs 6 and 7, respectively. The TGA results indicate that the Fe$_3$Ni catalyst only loses less than half a percent of its original mass, therefore, it is unlikely that the main deactivation pathway is due to coking.

The EDS of the spent Fe$_3$Ni sample shows small regions of higher Ni content, and to a lesser extent regions with higher Fe. However, in-situ XRD measurements do not reveal obvious agglomeration formation during reaction, and the absence of metal diffraction peaks suggests that the metal particles are most likely less than 2 nm in size (Supplementary Fig. 8). The Ni$_3$Pt catalyst loses about 8% of its original mass but does not illustrate signs of sintering. However, the coking over the Ni$_3$Pt catalyst at comparable propane conversion to Fe$_3$Ni is not significant.

**Oxidation states by in situ XANES.** In situ X-ray absorption near edge spectroscopy (XANES) measurements were conducted in order to identify the local environment of the metals under reaction conditions, as shown in Fig. 3. Additional details are available in Supplementary Note 3. The XANES data identified that under reaction conditions the Ni$_3$Pt catalyst consisted of metallic Pt (Supplementary Fig. 9) and that both the Fe$_3$Ni and Ni$_3$Pt catalysts consisted of metallic Ni (Fig. 3a). On the other hand, the Fe in the Fe$_3$Ni catalyst was in the oxidized form. The extended X-ray absorption fine structure (EXAFS) fitting of Fe$_3$Ni suggested the presence of an inserted oxygen through Fe–O–Fe as well as Fe–O bonds (Supplementary Table 4). Theofanidis et al. and Kim et al. studied DRM over higher loading Ni–Fe catalysts (8 wt.% Ni-5 wt.% Fe, and 8.8 wt.% Ni-2.1 wt.% Fe, respectively) supported on magnesium aluminate and they also observed oxidized Fe under in situ conditions but in an oxidation state of 2+[26,27]. For the Ni$_3$Pt catalyst, the EXAFS fitting indicates that the coordination number of the Pt–Pt and Pt–Ni bonds is 3.4 and 6.4, respectively, confirming the formation of the Pt–Ni bimetallic bond.

**Reaction pathways and DFT calculations.** Density functional theory (DFT) calculations were performed on bulk-terminated-Fe$_3$Ni(111) and Pt-terminated-Ni$_3$Pt(111) surfaces (Supplementary Fig. 10) to further gain insight into the potential reaction pathways for the oxidative C–H and C–C bond cleavage of propane to form *CH$_3$CHCH$_2$+H$_2$O(g) and *CH$_3$CH$_2$+*CO+H$_2$O(g), respectively. In these calculations, the surfaces are first modified by *O atoms assuming that *CO$_2$ dissociates to form *CO + *O. The DFT optimized geometries in Supplementary Fig. 11 show that the intermediates *CH$_3$CH$_2$CH$_2$O, *CH$_3$CH$_2$CHO, and *H$_2$O interact with the surfaces via the oxygen atoms while other intermediates

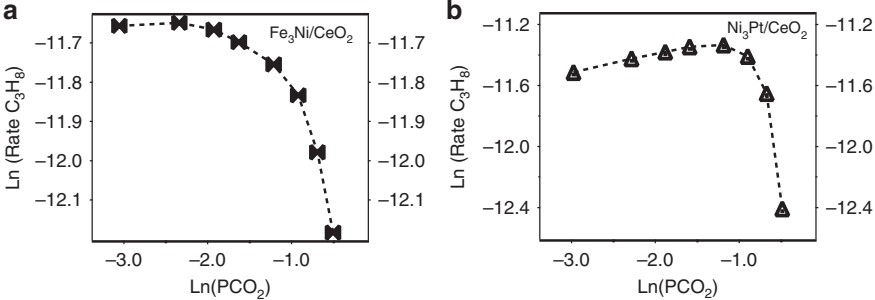

**Fig. 2** Effect of $CO_2$ partial pressure on the propane production rate. Plots for **a** Fe₃Ni and **b** Ni₃Pt. Total system pressure is 1 atm

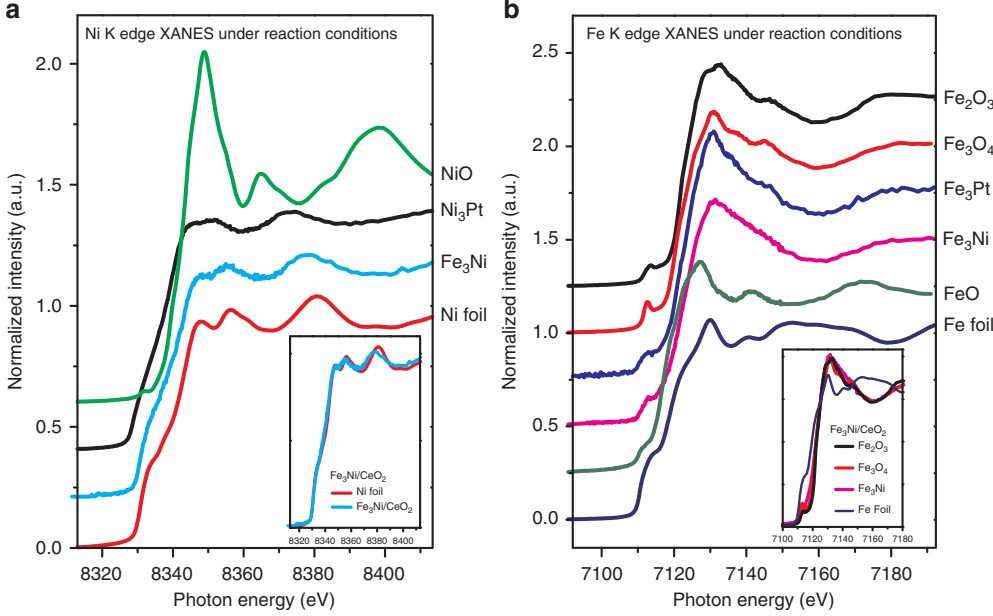

**Fig. 3** In-situ XANES spectra. **a** Ni and **b** Fe K edges of all the bimetallic catalysts with respective references. The insets show more detailed comparison of Fe₃Ni with model compounds

\*$CH_3CH_2CH_2$, \*$CH_3CHCH_2$, \*$CH_3CH_2$, and \*CO interact with the surfaces via the carbon atoms. It is noted that, even though the binding configurations of intermediates are similar on both surfaces, all the intermediates bind more strongly on bulk-terminated-Fe₃Ni (111) than on Pt-terminated-Ni₃Pt(111) (Supplementary Tables 5 and 6). The DFT calculated binding energies were then used to calculate the change in energy for the oxidative C–H and C–C bond scission of propane. On bulk-terminated-Fe₃Ni(111), Fig. 4a shows that the pathway for the oxidative C–H bond cleavage lies lower in energy than that for the C–C bond scission. In contrast, as shown in Fig. 4b on Pt-terminated-Ni₃Pt(111), the pathway for the C–C bond cleavage lies lower in energy than that for the C–H bond.

Overall the DFT results reveal that the C–C bond cleavage pathway is preferred on Ni₃Pt(111), while bulk-terminated-Fe₃Ni (111) favors the C–H bond cleavage pathway. Kinetically, this is also the case based on the comparison of activation energies (Supplementary Table 7). According to the DFT calculations, on Pt-terminated-Ni₃Pt(111), the \*O insertion reaction (\*$CH_3CH_2CH_2$ + \*O → \*$CH_3CH_2CH_2O$ + \*) along the C–C bond cleavage pathway ($\Delta E = -0.75$ eV and $E_a = 1.07$ eV) is thermodynamically and kinetically more favorable than the oxidative dehydrogenation reaction (\*$CH_3CH_2CH_2$ + \*O → \*$CH_3CHCH_2$ + \*OH) along the C–H bond cleavage pathway ($\Delta E = -0.51$ eV and $E_a = 1.33$ eV). In contrast, on bulk-terminated-Fe₃Ni(111), the oxidative dehydrogenation reaction ($\Delta E = 0.29$ eV and $E_a = 1.02$ eV) is more favorable than the \*O insertion reaction ($\Delta E =$

0.43 eV and $E_a = 3.30$ eV). These DFT predictions are in agreement with experimental observations, suggesting that the bulk-terminated-Fe₃Ni(111) surface promotes the oxidative C–H bond cleavage of propane to form \*$CH_3CHCH_2$ while the Pt-terminated-Ni₃Pt(111) surface promotes the C-C bond cleavage of propane to form \*CO.

To account for the potential FeO–Ni interfacial active sites based on the in situ experimental observation of oxidized Fe in the Fe₃Ni catalyst, further DFT calculations were carried out to investigate the pathways for the oxidative C–H and C–C bond cleavage of propane on the FeO/Ni(111) interface. For the $FeO_x$ clusters supported on Ni(111), both Fe₆O₉ and Fe₃O₃ clusters on three layer 7 × 7 Ni(111) and 5 × 5 Ni(111) surfaces (Supplementary Fig. 12) were considered. The oxygenated species (\*O, \*CO, \*$CH_3CH_2CH_2O$, \*$CH_3CH_2CHO$, and \*$CH_3CH_2CO$) prefer to adsorb at the interfacial sites while \*$C_xH_y$ species (\*$CH_3CH_2CH_2$, \*$CH_3CHCH_2$, and \*$CH_3CH_2$) most favorably adsorb on Ni(111) sites (Supplementary Table 8 and Supplementary Fig. 13). The energy diagram in Fig. 4c, calculated based on the DFT obtained binding energies of the potential intermediates, show that the first steps in oxidative C–C and C–H bond cleavage pathways are competitive. The subsequent step to form \*$CH_3CHCH_2$ is downhill in energy along the oxidative C–H bond cleavage pathway; in contrast, the subsequent steps are uphill in energy along the oxidative C–C bond cleavage pathway. Again, such thermodynamic predictions are fully supported by the calculated

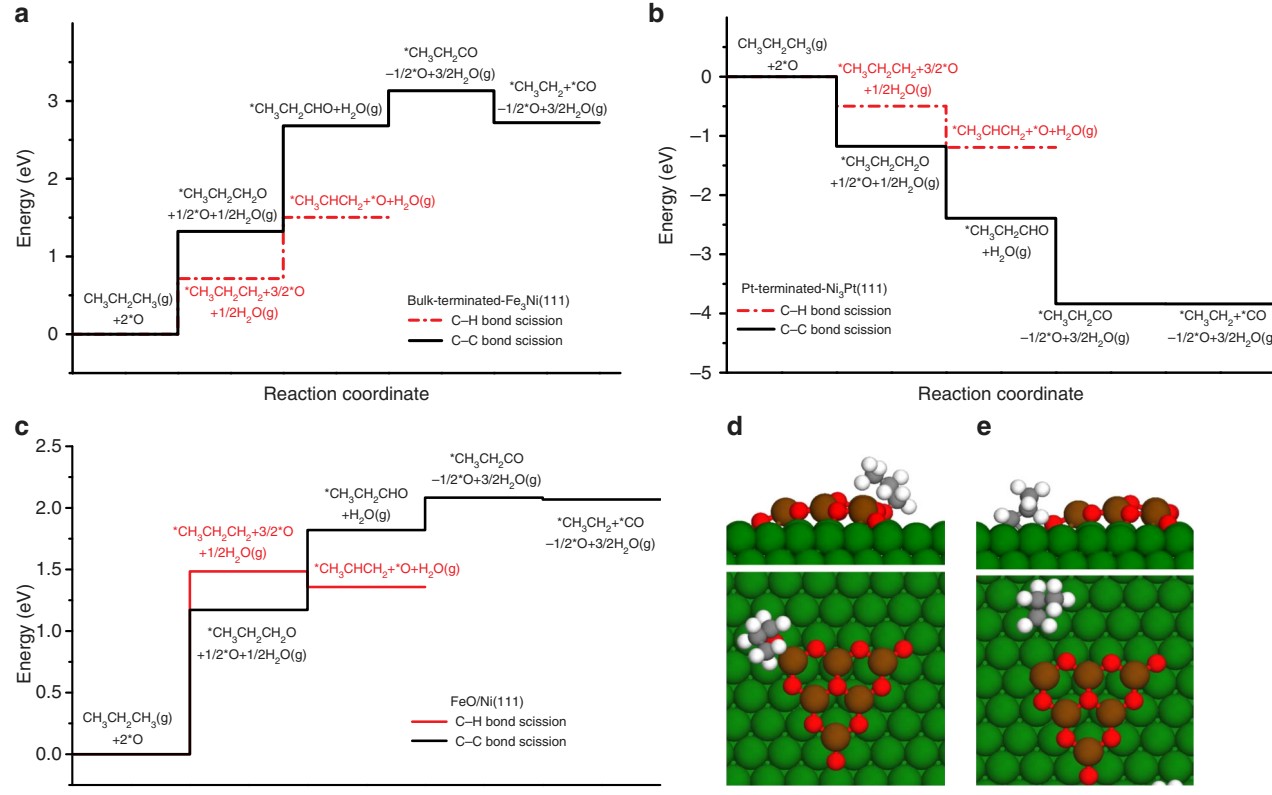

**Fig. 4** DFT calculated energy profiles for the oxidative C–H and C–C bond scission pathways. **a** Bulk $Fe_3Ni(111)$ surface, **b** Pt-terminated $Ni_3Pt(111)$ surface, and **c** FeO/Ni(111) interface as well as the optimized geometries of **d** $CH_3CH_2CH_2O$ and **e** $CH_3CH_2CH_2$ on FeO/Ni(111)

$E_a$, showing that the oxidative dehydrogenation reaction ($\Delta E = -0.40$ eV and $E_a = 0.29$ eV) is highly favorable over the *O insertion reaction ($\Delta E = 0.01$ eV and $E_a = 2.13$ eV) on the $Fe_2O_3/Ni(111)$ surface. This indicates that the oxidative dehydrogenation pathway should be more favorable than reforming, consistent with experimental observation.

Finally, on the three surfaces studied the desorption of *CO is expected to be a facile process due to the contribution of entropy at 823 K. $*C_2H_5$ is one of the reaction intermediates that undergoes O-insertion, C–H and C–C bond scission reactions to eventually produce CO and $H_2$. The *O species on Pt-terminated-$Ni_3Pt(111)$ react with $*C_xH_y$ to form the $*C_xH_yO$ intermediate, which promotes the C–C bond scission. In contrast, the more stable *O on bulk-terminated-$Fe_3Ni(111)$ and the FeO/Ni(111) interface are expected to remain on the surface, which facilitates the selective C–H bond scission of propane to produce propylene.

## Discussion

Overall, the oxidative dehydrogenation of propane with $CO_2$ has the potential to combine two underutilized[4–6] reactants to produce propylene or syngas. Two types of bimetallic catalysts have been identified for the $CO_2 + C_3H_8$ system. The DFT calculation results indicate that the bulk $Fe_3Ni(111)$ surface and the FeO/Ni(111) interface should favor C–H bond scission for the $CO_2$-ODHP pathway, whereas the Pt-terminated $Ni_3Pt(111)$ surface should favor the C–C bond cleavage for the DRP pathway. Flow reactor results are consistent with the DFT calculations as it was observed that the $Fe_3Ni$ catalyst is selective for propylene production, while the $Ni_3Pt$ catalyst shows good activity and CO selectivity. The oxidation states of the different metals provided by in situ XANES measurements reveal that $Fe_3Ni$ consists of oxidized Fe and metallic Ni. Future efforts should be geared toward enhancing propylene yield through the discovery of more stable and selective catalytic materials.

## Methods

**Density functional theory calculations**. Spin polarized[28,29] density functional theory (DFT) calculations were performed as an attempt to elucidate the possible pathways of C–C and C–H bond cleavage of propane over $Fe_3Ni(111)$, $Ni_3Pt(111)$ surfaces, and FeO/Ni(111) interface using the Vienna Ab Initio Simulation Package (VASP) code[30,31]. Projector augmented wave potentials were used to describe the core electrons with the generalized gradient approximation (GGA)[32,33] using PW91 functionals[34]. The Kohn–Sham one-electron wave functions were expanded by using a plane wave basis set with a kinetic energy cutoff of 400 eV. The Brillouin zone was sampled using a $3 \times 3 \times 1$ k-point grid in the Monkhorst–Pack scheme[35]. Ionic positions were optimized until Hellman–Feynman force on each ion was smaller than 0.02 eV/Å. The transition state of a chemical reaction was located using the climbing image nudged elastic band (CI-NEB) method implemented in VASP[36]. The activation energy ($E_a$) of a chemical reaction is defined as the energy difference between the initial and transition states while the reaction energy ($\Delta E$) is defined as the energy difference between the initial and final states.

**Catalyst preparation and flow reactor studies**. The catalysts evaluated in this study were synthesized through incipient wetness impregnation of metals onto commercially obtained $CeO_2$ (35–45 m²/g, Sigma-Aldrich). Flow reactor experiments were performed under atmospheric pressure utilizing a 1/4" quartz U-shaped reactor. All catalysts were reduced at 723 K for 1 h under a 1:1 $H_2$/Ar flow (40 mL/min total). Subsequently, the temperature was increased and held at 823 K in the presence of 1:1:2 $CO_2$, $C_3H_8$, and Ar for 12 h. Apparent activation barrier and reaction order experiments were conducted at slightly different reaction conditions to ensure operation in a true intrinsic kinetic regime and minimize transport effects. XANES measurements were conducted using a custom in situ micro-channel cell holding ~200 mg of catalyst (60–80 mesh) and a 4-channel vortex fluorescence detector.

**Data availability**. The data that support the findings of this study are available from the corresponding author upon request.

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

## Acknowledgements

The work is supported by the US Department of Energy (DOE) under contract number DE-SC0012704. The in situ XAS measurements were performed at the 2–2 beamline at the Stanford Synchrotron Radiation Lightsource (SSRL) at SLAC National Accelerator Laboratory (DE-AC02–76SF00515) and the 9-BM beamline of the Advanced Photon Source (APS) at the Argonne National Laboratory (DE-AC02-06CH11357). The DFT calculations were performed using computational resources at the Center for Functional Nanomaterials at BNL, a DOE Office of Science User Facility, and at the National Energy Research Scientific Computing Center (NERSC), a DOE Office of Science User Facility, supported by the Office of Science of the DOE under contract DE-AC02-05CH11231. E. G. acknowledges the US National Science Foundation Graduate Research Fellowship Program: DGE-16-44869.

## Author contributions

E.G. performed all flow reactor experiments and analyzed the results. S.K. and P.L. performed DFT calculations. B.Y. and S.Y. assisted E.G. at beamlines to collect in situ XAS data as well as perform data analysis. E.G. and J.G.C. prepared the manuscript and other authors made comments/additions.

## Additional information

**Competing interests:** The authors declare no competing interests.

