## [Peer Review File(PDF 2247 kb) · Nature Communications]

PEER REVIEW FILE

Reviewers' comments:

Reviewer #1 (Remarks to the Author):

The present manuscript aims to provide insight into the selectivity preferences of ceria-supported bimetallic catalysts (Fe, Ni, Pt) in CO₂-PDH or DRP reactions. In the introduction it is argued that ceria was selected as a support because it could store and release oxygen. The authors start fast forward in the Discussion section with flow catalytic results giving no information on how the catalysts were prepared, characterising the particle sizes of the active phases, etc. For example, what is the particle size of the catalytic systems in discussion? Next, the TOF values (Table 1) are presented for tested catalysts but no actual data on results of CO chemisorption experiments (except describing the general experimental set up in the ESI) is given. This should be provided. Results with monometallic Fe and Ni catalysts are discussed in the text but are lacking in Table 1.

It would be interesting to see XAS data for the first ca. 100 min on stream: what changes in the spectra accompany catalyst deactivation? Also a more detailed EXAFS study of the Ni₃Pt catalyst would have been helpful. Does Ni and Pt form an alloy? It is also not clear to me after how many hours of TOS was the data in figure 3 acquired? Considering that the authors acquired in-situ XAS it should be possible to track changes, e.g. in the oxidation states of Ni and Pt, with TOF and try to correlate this with changes in the materials activity. Additionally, the deactivation mechanism put forward for Fe₃Ni, viz sintering, is not very well supported by TEM/EDS. Are there additional deactivation mechanisms possible? Additional XRD data might also help here.

Concerning the DFT model, what is the rationale behind choosing these particular surface terminations? DFT models do not include ceria, even though the authors emphasised its role in oxygen storage on catalysis. How would the conclusions change if a more realistic model was considered that included ceria? While modelling an iron-rich catalyst, Fe₃Ni where Fe is in its oxidized form, why was FeO/Ni(111) chosen as a model, with only 6 Fe atoms on Ni(111) surface, and not the other way around?

Other notes: the catalysts coding needs some a clarification as labels Fe₃ and Ni₁ look

confusing, especially given the contradicting comment that “...respective monometallics (0.4% Ni and 0.2% Fe)..” What does labels 3 and 1 stand for in monometallic catalysts?

Reviewer #2 (Remarks to the Author):

The manuscript of Gomez et al. describes two new catalysts for the oxidative dehydrogenation of propane with CO₂, a Ni-Fe and Ni-Pt alloy that performs better than the constituting pure metals. As far as I am aware of, several groups try to develop catalysts for this potentially promising process, but none has published yet on this catalyst for propane dehydrogenation. The authors have published previously on similar, though not identical, catalysts for ethane and butane oxidative dehydrogenation by CO₂ (JCat & ACS Cat). Due to the novelty of the catalyst, the results of the communication are certainly useful to the field and may accelerate catalyst development for this process. Referencing is limited but seems to be appropriate.

The experimental part of the work seems to be well done and supported by the results, and involves traditional testing of catalyst activity/selectivity as well as advanced catalyst characterization. The DFT part is however the weaker part of the work: though the calculations are performed adequately, and the reported DFT results indeed support the experimental findings, the fact that the data support experiment is probably only so because the DFT analysis is incomplete. The DFT work considers only the electronic energies of all intermediates: reaction barriers have not been calculated though this is standard good practice these days. Evans-Polanyi based reasoning can of course be used for a rough estimate of the activation barrier – sometimes one has to approximate - but the authors should be more careful then to draw their conclusions.

There are also other comments to be made on the DFT work:

- The authors derive the model surfaces from XANES data, which is a very good starting point, indicating reduced Ni and oxidized Fe. However, the 2 catalyst models Fe₃Ni(111) and FeO/Ni(111) are limiting cases, and the actual catalyst surface may lay in between. Nothing is mentioned on the matter, and the surface energies of both states are not compared. Furthermore, will (111) still be the dominating facet in the case of an FeO top layer? And are the results sensitive to the geometry/size of the FeO ‘top layer’, since this is currently modelled as a 6-Fe-atom cluster on top of Ni(111)?

- There is no discussion at all on how the adsorption sites to which the intermediates bind have been identified, though there are many different possible sites, particularly for the Fe₃Ni and FeO/Ni models. This should at least be briefly discussed in the Supporting Information: default geometry optimization will not automatically evaluate all possible sites. The reader can currently not know that the identified states really are the minimum energy states.

- From figure 4 is derived that Fe₃Ni is more selective for propylene, while Ni₃Pt is more active

and CO-selective. This is indeed a likely possibility based on the data shown. However, the discussion of Figure 4 is more positive than the Figure actually looks like. Firstly, the schemes go up to 3 eV and down to -4 eV, which is enormous (and barriers are not even included, which will make them even higher in energy). Most likely, both schemes lead to complete inactivity because of too high barriers (for a and c) or poisoning by ethyl, CO and/or O species (for b). Secondly, the first step for (a) is propane adsorption/activation with a ΔE of +0.8 to +1.2 eV, depending on the path. In terms of ΔG this will be even higher by about 0.9 eV at 873 K, which will lead to negligibly low coverages. And finally, because the final gas phase states are not included in the plot (with gas phase O, CO, ...) there is no final check on the correctness of the DFT data. Trying to interpret the current data, one can only come up that in case of (a) and (c) energy will be released upon the desorption of ethyl, CO and O - which is unlikely - or these species will never desorb at all in case of (b), depending where the final gas phase states are.

As a minor comment, it is confusing that the cited selectivities in the text (e.g. page 4, line 75) are not the same as those in Table 1. I realize the difference (250 minutes on stream vs. averaged out between 10 to 12 hours on stream), but it is confusing.

Summarized, I endorse publication of this manuscript, but would suggest at least a more careful representation and interpretation of the DFT data, and preferably also suggest the calculation of reaction barriers - at least of the steps that are considered decisive for the determination of the selectivities.

Reviewer #3 (Remarks to the Author):

The manuscript by Gomez et. al proposes the use of Ni-based bimetallic catalysts for CO₂-based ODH or propane dry reforming processes to produce propylene or syngas while consuming CO₂. The use of CO₂ instead of O₂ as an oxidant is described to have multiple benefits: (1) The consumption of a greenhouse gas that is typically a waste stream, (2) a shift of dehydrogenation reaction thermodynamics to allow for higher conversion of the alkane at lower reaction temperatures than non-oxidative dehydrogenation or conventional dry reforming operations, and (3) the milder oxidizing power of CO₂ may reduce unselective reactions. The authors present NiFe/CeO₂ as a primarily dehydrogenation catalyst and NiPt/CeO₂ as a reforming catalyst, indicating that the close interaction of the two metals plays a role in determining process selectivity.

The most important contribution of this study was the attempt to modify nickel-based systems via bimetallic interactions to tune reaction selectivity to either dehydrogenation or reforming processes. This approach would provide valuable insight on how different metal interactions affect reactivity to the broader catalysis community. Beyond the product selectivity, the

bimetallic interactions may change catalyst stability, which is also an important factor for future development. This contribution, however, shows several shortcomings that limit its impact and need to be addressed for a more complete study.

First, the prepared catalysts appear to systematically replace one metal with another to explore their effect, but the FeNi and NiPt catalysts are different not only in their metal composition but also their atomic ratios. The authors should maintain the Ni:Metal ratio constant, which is not the case in the current work as we see Fe:Ni= 3:1 and Pt:Ni = 1:3. This is particularly important in the Fe₃Ni₁ sample, which is shown by EXAFS to possess Fe-O-Fe bonds. This result, however, may be in part due to the larger Fe motifs caused by higher Fe loadings compared to Ni. If Fe dopes a Ni particle (in contrast to Ni doping an Fe particle) would the EXAFS still show a similar structure? Without similar compositions, it becomes complicated to compare bimetallic catalysts, especially at the low loadings used in the manuscript.

Comparison of the catalyst via catalytic testing present NiFe as a dehydrogenation catalyst while NiPt is a reforming catalyst. Yet it is difficult to compare both Ni-based catalysts in this way (as seen in Table 1) when the product selectivities are not shown at iso-conversion. Judging by the fact that the NiFe catalyst has a C₃H₆ selectivity of about 60% at 2.7% C₃H₈ conversion, it is reasonable to expect this selectivity to decrease when testing this catalyst at the same conversion shown for NiPt. While I understand it is difficult to compare catalysts with different rates of deactivation, such an important claim of the work clearly requires a better comparison set. Any claims of product distribution differences must be done at isoconversion to be valid.

Furthermore, the authors should show conversion-selectivity trends to illustrate the differences between the catalysts and determine primary reaction products from their selectivity extrapolated to zero conversion (i.e. the y-intercept of selectivity in a conversion-selectivity plot). These selectivity trends would further validate their claims of CO being produced from reforming activity rather than simply overoxidation processes that CO₂ may ameliorate. I expect the authors must have this data readily available. All these considerations seek to answer unequivocally what are the main reaction pathways involved with these bimetallic materials. The NiFe catalyst in particular, has what amounts to almost 60-40 distribution of C₃H₆ and CO selectivities, which is hardly comparable to Cr-based catalysts for CO₂ – ODHP which show selectivities well above 80% towards propylene. I would be more compelled to agree with their claims if they presented more evidence to show that the produced CO does not come from reforming activity.

The kinetic experiments carried out (reaction order and apparent activation energy), while properly carried out, seem redundant or not used to complement the rest of their presented work. The authors highlight the competitive adsorption of C₃H₈ and CO₂ by comparing the effect of one substrate on the rate of consumption of the other (i.e. the effect of PCO₂ on rateC₃H₈) but they do not show the effect of PC₃H₈ on rateC₃H₈, for instance. In conventional ODHP with oxygen, there is a first order dependence on PC₃H₈ due to the weak adsorption of the alkane on the active site, however I would not expect this to be the case if the adsorption-like dependence shown in Figure S2-2 holds. Complementing this figure with the other reaction order

experiments would present a more complete picture with which to develop an overall rate equation, for instance. Similarly, the activation energy experiments only present the activation energy of the purportedly primary reaction pathway on each catalyst (Figure S2-1). To complement these, the activation energy of the minor pathway (i.e. reforming with the NiFe catalyst) should be presented. If the authors' hypotheses are valid, the activation energies should mirror the results of their DFT calculations however, as discussed below, the DFT calculations cannot be directly compared due to the lack of energy barriers calculated.

Related to the experimental reactivity results, the authors present DFT calculations that suggest that the NiFe and FeO-Ni surfaces show favorable energy pathways for dehydrogenation. These claims, however, are based on adsorption enthalpies, which represent only part of the overall picture of a chemical reaction. Any claim to compare both C-H and C-C bond breaking requires activation energy calculations, which would also complement the apparent energy barriers observed in experiment and inform of the kinetics of these two reactions. It is unclear to me why energy barriers would not be included in such study. Furthermore, the DFT calculations show that the NiFe and FeO-Ni surface could favor C-H bond breaking but the authors never discuss why there is such a significant amount of CO still being formed with the NiFe catalyst, which I would presume is due to the unselective overoxidation of the olefin products.

Other minor issues to address:

- The TGA analysis presented in Figure S5-1 aims to show that the NiFe catalyst does not coke significantly, but the comparison is not exactly fair as the two catalysts were run at significantly different conversions. The catalysts should be run at comparable levels of conversion and then their mass change compared. While the result may be the same as presented, the experimental approach needs to be improved. Furthermore, it is unclear why the figure itself is presented with two temperature axes increasing in opposite directions of the x-axis. This approach is confusing and may lead to misinterpretation. If the authors wish to prevent overlap of the profiles, they should use two figures instead.
- The EXAFS fitting shown in Figure S10-1 seems incomplete, as the Ni fit is only reasonable at the 2Å distance while the rest of the experimental data does not agree with the fit. I would expect proper fitting up to at least 3Å for any valid bond distance claims to be made. The Fe edge is somewhat better but still should be improved. If the authors explained how they approached their fitting procedure, the reader may be able to understand why the fits shown are deemed valid when intuitively they seem incomplete.

Based on the incomplete reactivity studies, different catalyst compositions and lack of energy barriers/transition states in the DFT portion of the work I cannot recommend this manuscript for publication in a high-impact publication such as Nature Communications. Furthermore, the catalytic performance that is presented shows dehydrogenation activity far below the currently state of the art and no inherent benefit of these materials seems to be proposed. The authors should significantly modify their experimental approach to improve this work to be publication-ready in another more specialized journal.

Combining CO₂ Reduction with Propane Oxidative Dehydrogenation over Bimetallic Catalysts

Elaine Gomez¹, Shyam Kattel², Binhang Yan², Siyu Yao², Ping Liu², and Jingguang G. Chen^{1,2*}

Response to Reviewers

Reviewer #1:

(1) The present manuscript aims to provide insight into the selectivity preferences of ceria-supported bimetallic catalysts (Fe, Ni, Pt) in CO₂-PDH or DRP reactions. In the introduction it is argued that ceria was selected as a support because it could store and release oxygen. The authors start fast forward in the Discussion section with flow catalytic results giving no information on how the catalysts were prepared, characterising the particle sizes of the active phases, etc. For example, what is the particle size of the catalytic systems in discussion?

[Response]

Due to the format of the manuscript, the *Experimental Methods* section is located after the discussion. The description regarding the catalyst synthesis procedure is on line 241 and more detailed information is available in the supporting information section *SI*. All catalysts were synthesized via incipient wetness impregnation of metals onto commercially obtained CeO₂ (35-45 m²/g, Sigma Aldrich). Metal precursor salts Pt(NH₃)₄(NO₃)₂, Ni(NO₃)₂·6H₂O, and Fe(NO₃)₃·9H₂O (Alfa Aesar) were dissolved in a volume of water equivalent to the pore volume of an aliquot sample of the support. The metal salt solutions were added by dropwise addition to the support and subsequently mixed. The catalysts were dried at 363 K for 6 hours in order to remove water and to allow the salt to crystallize on the pore surface. Once dried, the catalysts were calcined at 563 K for 2 hours.

Regarding the metal particle size, our group has previously determined that the average particle size of reduced Ni₃Pt/CeO₂ is 2.3 nm from TEM measurements^[1]. However, it is very challenging to obtain a reliable particle size distribution for Fe₃Ni/CeO₂ using TEM because of the difficulties in measuring 3d elements (Fe and Ni) over the CeO₂ support. As will be discussed later in response to Comment 7 by Reviewer 1, the in-situ XRD measurements Fe₃Ni/CeO₂ did not show any diffraction patterns due to metal particles, suggesting that the particle size of Fe₃Ni should be less than ~2 nm.

[Action]

The following sentence has been added on line 70 of the manuscript to address catalyst preparation before the *Experimental Methods* section:

“All catalysts were synthesized via incipient wetness impregnation of metals onto commercially obtained CeO₂ (35-45 m²/g, Sigma Aldrich). For additional details see *Experimental Methods* or *SI* section *SI*.”

[1] B. Yan, X. Yang, S. Yao, J. Wan, M. N. Z. Myint, E. Gomez, Z. Xie, S. Kattel, W. Xu, J. G. Chen, *ACS Catal.* **2016**, *6*, 7283–7292.

Reduced particle sizes are now mentioned in the SI section *SI* as,

“The average metal particle size of reduced Ni₃Pt/CeO₂ was determined to be 2.3 nm from TEM measurements in a previous study ^[1]. However, it is very challenging to obtain a reliable particle size distribution for Fe₃Ni/CeO₂ using TEM because of the difficulties in measuring 3d elements (Fe and Ni) over the CeO₂ support. From the in-situ XRD measurements (Figure S6.2-1), Fe₃Ni/CeO₂ did not show any diffraction patterns from metal particles, suggesting that the particle size of Fe₃Ni should be less than ~2nm.”

(2) Next, the TOF values (Table 1) are presented for tested catalysts but no actual data on results of CO chemisorption experiments (except describing the general experimental set up in the ESI) is given. This should be provided.

[Response]

In order to compare catalysts according to the available active sites determined by CO chemisorption, turnover frequency (TOF) values are provided in Table 1, which begins on line 99 of the main text. The respective CO uptake values for Fe₃Ni, Fe₃Pt, and Ni₃Pt are listed in the second column of Table 1 following values for conversion, TOF, selectivity, and yield.

[Action]

The following modification has been added to the sentence on line 70:

“Flow reactor studies measuring both CO₂-ODHP and DRP activity simultaneously are summarized in Table 1 along with CO chemisorption values.”

Additionally, the CO chemisorption results are also now listed in the supporting information section *S3* in Table S3-1.

Table 1. Bimetallic catalyst flow reactor results for CO₂ + C₃H₈ reaction (10 mL/min each) at 823 K with Ar diluent (20 mL/min) and 100 mg of catalyst (16-20 mesh). Values are obtained by averaging data from hours 10-12. Selectivity and yield are on a C₃H₈ basis (including only carbonaceous species). Catalysts are synthesized by atomic ratios corresponding to a 1.67 wt.% Pt₁ basis, thus the weight percent of Fe₃, Ni₁, and Ni₃ are 1.5, 0.5, and 1.5, respectively.

Catalyst /CeO ₂	CO Uptake μmol·g ⁻¹	Conversion (%)		TOF (site ⁻¹ ·min ⁻¹)		Selectivity (%)					Yield (%)	
		CO ₂	C ₃ H ₈	CO ₂	C ₃ H ₈	CO	C ₃ H ₆	CH ₄	C ₂ H ₆	C ₂ H ₄	CO	C ₃ H ₆
Fe ₃ Ni	31.9	4.0	2.7	5.7	3.4	40.2	58.2	0.8	0	0.8	1.1	1.6
Fe ₃ Pt	31.5	2.6	1.1	3.5	1.5	65.1	32.0	1.3	0	1.6	0.7	0.3
Ni ₃ Pt	50.1	39.4	11.6	37.5	10.5	96.2	2.8	0.83	0.1	0	11.1	0.3

(3) Results with monometallic Fe and Ni catalysts are discussed in the text but are lacking in Table 1.

[Response]

Key comparisons among bimetallic and respective monometallic catalysts are discussed in the *Results* section of the manuscript. Table 1 on line 99 compares the activity of the three bimetallic systems tested for the reactions of CO₂+C₃H₈, whereas the data regarding the comparison among respective monometallic catalysts were given in the supporting information section S7-2. On line 79 the reader is referred to Table S7-2 to view the differences in conversion and yield among Fe₃Ni, Ni₁, and Fe₃. Similarly, on line 86 the reader is prompted to Figure S7-3 which illustrates selectivity, conversion, and normalized CO₂ conversion for Ni₃Pt, Ni₃, and Pt₁.

[Action]

To facilitate viewing monometallic data, the results are now listed in Table 1 of the manuscript. Fe₃ is inactive at the tested conditions and therefore not included in Table1, and is mentioned in the manuscript on line 76.

Modified Table 1. Catalyst flow reactor results for CO₂ + C₃H₈ reaction (10 mL/min each) at 823 K with Ar diluent (20 mL/min) and 100 mg of catalyst (16-20 mesh). Catalysts marked with an asterisk indicate that the sample was diluted to achieve comparable C₃H₈ reactant conversion to Fe₃Ni. Values are obtained by averaging data from hours 10-12. Selectivity and yield are on a C₃H₈ basis (including only carbonaceous species). Catalysts are synthesized by atomic ratios corresponding to a 1.67 wt.% Pt₁ basis, thus the weight percent of Fe₃, Ni₁, and Ni₃ are 1.43, 0.5, and 1.5, respectively. The nomenclature assigned by subscripts such as in Fe₃Ni means that there are 3 atoms of Fe for every atom of Ni.

Catalyst /CeO ₂	CO uptake μmol·g ⁻¹	Conversion (%)		TOF (site ⁻¹ ·min ⁻¹)		Selectivity (%)					Yield (%)	
		CO ₂	C ₃ H ₈	CO ₂	C ₃ H ₈	CO	C ₃ H ₆	CH ₄	C ₂ H ₆	C ₂ H ₄	CO	C ₃ H ₆
Fe ₃ Ni	31.9	4.0	2.7	5.7	3.4	40.2	58.2	0.8	0	0.8	1.1	1.6
Fe ₃ Pt	31.5	2.6	1.1	3.5	1.5	65.1	32.0	1.3	0	1.6	0.7	0.3
Ni ₃ Pt	50.1	39.4	11.6	37.5	10.5	96.2	2.8	0.83	0.1	0	11.1	0.3
*Ni ₃ Pt	-	7.8	2.2	-	-	87.8	11	0.9	0.0	0.3	2.0	0.2
Ni ₁	13.1	9.3	3.0	31.9	8.9	86.8	12.3	0.60	0.24	0	2.6	0.4
Ni ₃	37.7	32.8	9.6	40.2	11.4	94.9	2.9	2.11	0.05	0.06	9.1	0.3
Pt ₁	16.0	4.2	1.6	8.1	2.8	77.0	21.2	0.8	0.9	0	1.3	0.4

Table S7-2: Comparison among Fe₃Ni₁/CeO₂ and the respective monometallics for the reaction of CO₂ + C₃H₈ (10 mL/min each) with Ar diluent (20 mL/min) at 823 K and 100 mg of catalyst. Steady state conversions of CO₂ and C₃H₈ as well as the C₃H₆ and CO yields are included.

Catalyst Supported on CeO ₂	Conversion (%)		Yield (%)	
	CO ₂	C ₃ H ₈	C ₃ H ₆	CO
Fe ₃ Ni ₁	4.0	2.7	1.6	1.1
Ni ₁	9.3	3.0	0.4	2.6
Fe ₃	0.10	0.45	0.2	0.0

Figure S7-3: Comparison among Ni₃Pt₁/CeO₂ and the respective monometallics for the reaction of CO₂ + C₃H₈ (10 mL/min each) with Ar diluent (20 mL/min) at 823 K and 100 mg of catalyst. (a) Selectivity and conversion of Ni₃Pt, Ni₃, and Pt₁ as well as (b) normalized CO₂ conversion vs. time on stream.

(4) It would be interesting to see XAS data for the first ca. 100 min on stream: what changes in the spectra accompany catalyst deactivation?

[Response]

We agree with the Reviewer that the in-situ X-ray absorption fine structure spectra (XAS) data for the first 100 minutes of time on stream (TOS) can show valuable structural information for active catalysts. Precisely so, in-situ measurements were taken in the range of 10-130 minutes on stream and the final spectra is a merge of multiple scans, each within itself a sum of multiple states. The X-ray absorption near edge structure (XANES) region reflects the initial 5 min state, while the extended X-ray absorption fine structure (EXAFS) region is an average from the 5-20 min state. Due to limitation in time resolution at the Stanford beamline, we were only able to

measure three sets of XANES and EXAFS for each metal edge within 10-130 minutes on stream. Based on Reviewer's comments we compared the three individual spectra, as shown in Figure S10-A below. We did not observe any notable differences between the first, second, and third scan of any given edge in question. Because XAS is not an inherently surface sensitive technique, in this instance it is difficult to correlate initial deactivation to the in-situ scans as it may be possible that initial deactivation is influenced by changes occurring on the catalyst surface.

Figure S10-A: In-situ reaction scans of Fe_3Ni and Ni_3Pt with indication of time on stream (TOS) for the three scans taken at each edge energy.

[Action]

Figure S10-A has been added to the supplementary information section *S10*.

(5) Also a more detailed EXAFS study of the Ni_3Pt catalyst would have been helpful. Does Ni and Pt form an alloy?

[Response]

Based on Reviewer's comment we have performed additional XANES and EXAFS measurements of $\text{Ni}_3\text{Pt}/\text{CeO}_2$ under in-situ conditions. The EXAFS fitting of Ni_3Pt indicates that

Pt is bonded to both Ni and Pt. The coordination number of the Pt-Pt and Pt-Ni bonds is 3.4 and 6.4, respectively, confirming the formation of the Pt-Ni bimetallic bond under reaction conditions.

[Action]

The in-situ XANES spectra for the Pt L_{III} edge, which illustrates that Pt is also metallic in the Ni₃Pt/CeO₂ catalyst, has been added to the SI section S10 in Figure S10-2. The EXAFS fitting table is also now available in section S10 and the following modifications to the manuscript have been made:

On line 133:

“The XANES data identified that under reaction conditions the Ni₃Pt catalyst consisted of metallic Pt (Figure S10-2) and that both the Fe₃Ni and Ni₃Pt catalysts consisted of metallic Ni (Figure 3a).”

On line 140:

“For the Ni₃Pt catalyst, the EXAFS fitting indicates that the coordination number of the Pt-Pt and Pt-Ni bonds is 3.4 and 6.4 (Table S10-2), respectively, confirming the formation of the Pt-Ni bimetallic bond.”

Table S10-2: Summary of EXAFS fitting analysis for the reaction of CO₂ + C₃H₈ over Ni₃Pt/CeO₂.

Sample/Reaction	Edge	Shell	R (Å)	C.N.	σ^2
Ni ₃ Pt/CeO ₂	Pt L _{III}	Pt-Ni	2.56+/-0.02	6.4+/-1.4	0.010(2)
C ₃ CO ₂ -ODH	Pt L _{III}	Pt-Pt	2.78+/-0.03	3.4+/-0.8	0.010(2)

Figure S10-2: *In-situ* XANES spectra for the Pt L_{III} edge of Ni₃Pt/CeO₂ with respective references.

(6) It is also not clear to me after how many hours of TOS was the data in figure 3 acquired? Considering that the authors acquired in-situ XAS it should be possible to track changes, e.g. in the oxidation states of Ni and Pt, with TOF and try to correlate this with changes in the materials activity.

[Response]

In our studies, the in-situ XAS data collection began after the catalyst had been in contact with reaction gasses and at temperature for 10 minutes. Spectra were collected by alternating between edge energies, starting with the Ni K edge until there were three scans on each edge. The oxidation state of Ni was determined by merging the XANES region of the first (TOS 10-15 min), third (TOS 50-55 min), and fifth (TOS 90-95 min) scans. The XANES scans for Fe K or Pt L_{III} edge were taken at TOS 30-35, 70-75, and 110-115 min. No significant spectroscopic changes were detected on a given edge within the allotted time, as described in Figure S10-A for Comment 4 above. Figure S7-A below illustrates normalized TOF following time on stream after the reactor and GC lined out (81 min). From the three Ni scans for either catalyst and the three Fe K/Pt L_{III} edge scans it is not evident that there are any changes in oxidation that accompany the observed change in TOF. It is possible that initial changes of TOF with TOS are due to modifications of the catalyst surface, which make it difficult to detect using the relatively bulk sensitive XAS technique.

Figure S7-A: Normalized TOF for reactants CO₂ and C₃H₈ over Ni₃Pt/CeO₂ following time on stream after 81 minutes.

[Action]

The following sentences from the response above have been added to the SI section S10 and Figure S7-A is now available in section S7.

“In-situ XAS data collection began after the catalyst had been in contact with reaction gasses and at temperature for 10 minutes. Spectra were collected by alternating between edge energies, starting with the Ni K edge until there were three scans on each edge. The oxidation state of Ni was determined by merging the XANES region of the first (TOS 10-15 min), third (TOS 50-55 min), and fifth (TOS 90-95 min) scans. The XANES scans for Fe K or Pt L_{III} edge were taken at TOS 30-35, 70-75, and 110-115 min. No significant spectroscopic changes were detected on a given edge within the allotted time.”

(7) Additionally, the deactivation mechanism put forward for Fe₃Ni, viz sintering, is not very well supported by TEM/EDS. Are there additional deactivation mechanisms possible? Additional XRD data might also help here.

[Response]

Coking is a usual suspect for deactivation in reactions containing allylic hydrogens due to their lower bond dissociation energy compared to the saturated hydrocarbons. Thermogravimetric results, however, suggest that there is very little coke that burns off the spent Fe₃Ni/CeO₂ sample. Energy dispersive spectroscopy indicated that the reduced sample was well dispersed, but the spent sample showed regions of higher Ni content. Based on the Reviewer's comment, in-situ XRD measurements have been added to investigate metal sintering. The sample underwent reduction and reaction treatment at the same temperatures and in proportional gaseous environments as in steady-state experiments. The in-situ data reveal that there are no obvious metal phases, potentially a consequence of low loading. However, it is possible to infer that the metal particles cannot be larger than approximately 2nm, as anything above this would be observed in XRD measurements. Another potential cause for deactivation may be due to intermediates from propane with high binding energy occupying more catalytic sites as time progresses. This is consistent with the kinetics experiments that as the partial pressure of propane increases the rate of CO₂ and propane conversion eventually decline (supporting information Figure S2-2 and S2-3).

Figure S6-2-1: Fe₃Ni/CeO₂ in-situ XRD intensity vs. 2θ for reduction and reaction treatment.

[Action]

The in-situ XRD for Fe₃Ni/CeO₂ data has been added to the supporting information section S6-2. The manuscript has been modified to reflect the findings of the in-situ XRD measurements as follows,

“The EDS of the spent Fe₃Ni sample shows small regions of higher Ni content, and to a lesser extent regions with higher Fe. However, in-situ XRD measurements do not reveal obvious agglomeration formation during reaction, and the absence of

metal diffraction peaks suggests that the metal particles are most likely less than 2 nm in size (Figure S6-2-1).”

(8) Concerning the DFT model, what is the rationale behind choosing these particular surface terminations? DFT models do not include ceria, even though the authors emphasised its role in oxygen storage on catalysis. How would the conclusions change if a more realistic model was considered that included ceria?

[Response]

Two models were used to describe the Fe₃Ni cluster supported on CeO₂. One was the bulk terminated Fe₃Ni(111) surface using the L12 cubic crystal structure and a four layer 4 × 4 surface slab (Figure S9-1a). The other was the FeO_x clusters supported on Ni(111), where both Fe₆O₉ and Fe₃O₃ clusters on 3 layer 7 × 7 Ni(111) and 5 × 5 Ni(111) surfaces (Figure S9-1c) were considered. In this way, the experimentally observed oxidation of Fe on the surface under reaction conditions can be captured in DFT. Similarly, the Ni₃Pt cluster supported on CeO₂ was described by a Ni₃Pt(111) surface using the L12 cubic crystal structure and a four layer 4 × 4 surface slab (Figure S9-1b), where the formation of the Pt skin on the surface was included to consider the segregation of Pt according to our experimental observations.^[2] Ni₃Pt(111) and Fe₃Ni(111) surfaces were modeled since for such cubic structure, the (111) facet is the energetically most favorable low index surface.

In both cases, the effect of CeO₂ and the particle size was not taken into consideration. A more realistic model including CeO₂ would be very computationally demanding, especially considering that the average size of the PtNi nanoparticles is 2.3 nm in our experiment. For point of reference, it took us approximately two months to calculate the reaction network of propane oxidative dehydrogenation over the metal slabs without CeO₂. Similar calculations including a 2.3 nm particle and a CeO₂ substrate would take at least an order of magnitude more computation time. According to our experimental results, CeO₂ plays a crucial role in the initial CO₂ activation, while the subsequent reforming and dehydrogenation reactions take place on the surface of bimetallic particles. The different trend in the catalytic behavior between Fe₃Ni/CeO₂ and Ni₃Pt/CeO₂ observed experimentally during propane oxidative dehydrogenation is mostly associated with the surface of bimetallic alloys, which justifies our focus on the trend of different bimetallic surfaces in the DFT calculations.

[Action]

The following details about the DFT models have been included in section S9 of the SI:

“Two models were used to describe the Fe₃Ni cluster supported on CeO₂. One was the bulk terminated Fe₃Ni(111) surface using the L12 cubic crystal structure and a four layer 4 × 4 surface slab (Figure S9-1a). The other was the FeO_x clusters supported on Ni(111), where both Fe₆O₉ and Fe₃O₃ clusters on 3 layer 7 × 7 Ni(111) and 5 × 5 Ni(111) surfaces (Figure S9-1c) were considered. In agreement with our experimental observation, the DFT calculations showed that in the presence of oxygen from *CO₂ dissociation, Fe segregation is thermodynamically more

[2] W. W. Lonergan, D. G. Vlachos, J. G. Chen, *J. Catal.* **2010**, 271, 239–250.

favorable by -0.08 eV/atom than Ni segregation due to the stronger Fe-O bond than Ni-O. Such O-driven Fe segregation can result in the formation of FeO_x particles over the surface, resulting in the surface and subsurface layers being Fe-deficient or Ni-rich. Therefore, the selection of FeO/Ni(111) is a reasonable model to represent the FeO/Fe₃Ni(111) interface. The model of Fe₆O₉/Ni(111) is to describe the active interfacial sites. The choice of such small clusters is to achieve a compromise between the computational cost and a reasonable cluster size to explain the trends observed experimentally.

Similarly, the Ni₃Pt cluster supported on CeO₂ was described by a Ni₃Pt(111) surface using the L12 cubic crystal structure and a four layer 4 × 4 surface slab (Figure S9-1b), where the formation of the Pt skin on the surface was included to consider the segregation of Pt according to our experimental observations.^[2] In both cases, the effect of CeO₂ and the particle size was not taken into consideration. According to our experimental results, CeO₂ plays a crucial role in the initial CO₂ activation, while the subsequent reforming and dehydrogenation reactions take place on the surface of bimetallic particles. The different trend in the catalytic behavior between Fe₃Ni/CeO₂ and Ni₃Pt/CeO₂ observed experimentally during propane oxidative dehydrogenation is mostly associated with the surface of bimetallic alloys, which justifies our focus on the trend of different bimetallic surfaces in the DFT calculations.”

(9) While modelling an iron-rich catalyst, Fe₃Ni where Fe is in its oxidized form, why was FeO/Ni(111) chosen as a model, with only 6 Fe atoms on Ni(111) surface, and not the other way around?

[Response]

Based on Reviewer’s comments, additional DFT calculations were performed to study the surface segregation of Fe₃Ni(111) in vacuum and upon interaction with oxygen from CO₂ dissociation (*CO₂ → *CO + *O). We used a four layer 2 × 2 Fe₃Ni(111) surface slab with 0.25 ML oxygen coverage. On Fe₃Ni(111), Ni segregation is thermodynamically favorable (-0.09 eV/Ni atom), rather than Fe segregation (0.21 eV/Fe atom). Therefore, the Fe₃Ni(111) surface is likely Ni-rich. In the presence of oxygen, Fe segregation is thermodynamically more favorable by -0.08 eV/atom than Ni segregation due to the stronger Fe-O bond than Ni-O. Such O-driven Fe segregation can result in the formation of FeO_x particles over the surface, and the surface and subsurface layers are Fe-deficient or Ni-rich. Therefore, our model FeO/Ni(111) is a reasonable model to represent FeO/Fe₃Ni(111).

The model of Fe₆O₉/Ni(111) is to describe the active interfacial sites. The choice of such small clusters is to achieve a compromise between the computational time and a reasonable cluster size to account for the nanoparticles used experimentally. Increasing the FeO cluster size would require an even larger Ni(111) unit cell than 7 × 7, which will be too computational demanding for DFT calculations to describe such a complex reaction network. Therefore, we feel that our cluster size selection is justified for obtaining the trend among different catalysts while keeping the computation time to be in the timeframe of months instead of years.

[Action]

As described in the Action for the previous Comment 8, the following modifications has been made in the section S9 of the SI.

“In agreement with our experimental observation, the DFT calculations showed that in the presence of oxygen from *CO₂ dissociation, Fe segregation is thermodynamically more favorable by -0.08 eV/atom than Ni segregation due to the stronger Fe-O bond than Ni-O. Such O-driven Fe segregation can result in the formation of FeO_x particles over the surface, resulting in the surface and subsurface layers being Fe-deficiency or Ni-rich. Therefore, the selection of FeO/Ni(111) is a reasonable model to represent the FeO/Fe₃Ni(111) interface. The model of Fe₆O₉/Ni(111) is to describe the active interfacial sites. The choice of such small clusters is to achieve a compromise between the computational cost and a reasonable cluster size to explain the trends observed experimentally.”

(10) Other notes: the catalysts coding needs some a clarification as labels Fe₃ and Ni₁ look confusing, especially given the contradicting comment that “...respective monometallics (0.4% Ni and 0.2% Fe)..” What does labels 3 and 1 stand for in monometallic catalysts?

[Response]

The authors agree that further clarification regarding the atomic ratio and metal loading of the catalysts should be provided. The metal loadings of the catalysts are as indicated on the caption of Table 1, “Catalysts were synthesized by atomic ratios corresponding to a 1.67 wt.% Pt₁ basis, thus the weight percent of Fe₃, Ni₁, and Ni₃ are 1.5, 0.5, and 1.5, respectively.” Values were rounded on the table, but the exact values are now listed as 1.43 wt.% Fe₃, 0.5 wt.% Ni₁, and 1.50 wt.% Ni₃. An example calculation is shown below:

Example using 1.67 wt.% Pt as a reference:

$$\text{Weight \% of Ni}_1 = \frac{1.67}{MW \text{ of Pt}} \times MW \text{ of Ni} = 0.5$$

Thus, to obtain an equivalent amount of atomic concentration corresponding to 1.67 wt.% Pt, the Ni loading is 0.5 wt.%. If the catalyst is to have 3 atoms of Fe for every atom of Ni, then the Fe wt.% loading would be 1.43 wt.%.

Please note that the percentages on line 79 from the statement, “... respective monometallics (0.4% Ni and 0.2% Fe) ...” do not indicate the metal loading of each catalyst, but rather the yield of propylene from steady state flow reactor results.

[Action]

To clarify metal loading and the nomenclature, a sentence has been added to the caption of Table 1:

“The nomenclature assigned by subscripts such as in Fe₃Ni means that there are 3 atoms of Fe for every atom of Ni.”

The following corrections have been made to lines 79-80 to indicate that the written values correspond to propylene yield and not metal loading.

“The differences among the propylene yields on a C₃H₈ basis provided in Table S7-2 over Fe₃Ni (1.6% C₃H₆ yield) and the respective monometallics (C₃H₆ yield of 0.4% over Ni and 0.2% over Fe) indicate that there is a synergistic effect from the formation of the bimetallic Fe₃Ni catalyst.”

Reviewer #2:

The manuscript of Gomez et al. describes two new catalysts for the oxidative dehydrogenation of propane with CO₂, a Ni-Fe and Ni-Pt alloy that performs better than the constituting pure metals. As far as I am aware of, several groups try to develop catalysts for this potentially promising process, but none has published yet on this catalyst for propane dehydrogenation. The authors have published previously on similar, though not identical, catalysts for ethane and butane oxidative dehydrogenation by CO₂ (JCat & ACS Cat). Due to the novelty of the catalyst, the results of the communication are certainly useful to the field and may accelerate catalyst development for this process. Referencing is limited but seems to be appropriate. The experimental part of the work seems to be well done and supported by the results, and involves traditional testing of catalyst activity/selectivity as well as advanced catalyst characterization. The DFT part is however the weaker part of the work: though the calculations are performed adequately, and the reported DFT results indeed support the experimental findings, the fact that the data support experiment is probably only so because the DFT analysis is incomplete.

[Response]

We thank the Reviewer for the careful evaluation of our work. We have performed additional DFT calculations in the revised manuscript to address the comments. Please see below for the individual concerns.

(1) The DFT work considers only the electronic energies of all intermediates: reaction barriers have not been calculated though this is standard good practice these days. Evans-Polanyi based reasoning can of course be used for a rough estimate of the activation barrier – sometimes one has to approximate - but the authors should be more careful then to draw their conclusions.

[Response]

We agree with the Reviewer’s comments and have performed extensive DFT calculations that are described below. Following the Reviewer’s suggestion, we performed additional DFT calculations to estimate activation energies (E_a) on Pt-terminated-Ni₃Pt(111), bulk-terminated-Fe₃Ni(111) and at the FeO/Ni(111) interface. However, there are possibly over 100 elementary reactions and the DFT calculations of activation energy of all these steps is not feasible in such short time even using our relatively large computing cluster. Instead, only the key steps for the oxidative C-H and C-C bond cleavage pathways were selected for transition state calculations (Figure 4). In addition, to further save time the size of the FeO cluster was decreased from Fe₆O₉ to Fe₃O₃ to capture the trend in activation energies for cleaving the C-H and C-C bonds As illustrated in Figure 4, the energy diagrams using Fe₃O₃/Ni(111) show similar trends to those obtained on Fe₆O₉/Ni(111) (Figure S9-4), even though the smaller FeO cluster generally binds intermediates more strongly than the larger FeO cluster. Therefore, we believe that both

Fe₃O₃/Ni(111) and Fe₆O₉/Ni(111) are reasonable models to represent the FeO/Fe₃Ni(111) interface.

One of the key steps considered for transition state calculations is the activation of propane (i. e. *CH₃CH₂CH₃ + * → *CH₃CH₂CH₂ + *OH), which is a common step in both C-H and C-C bond cleavage (Figure 4) and directly affects the overall conversion. In addition, the dehydrogenation of *CH₂CH₂CH₂ (*CH₃CH₂CH₂ + *O → *CH₃CHCH₂ + *OH) and its reaction with *O to form of *CH₃CH₂CH₂O (*CH₃CH₂CH₂ + *O → *CH₃CH₂CH₂O + *) are also key steps to control the selectivity toward oxidative dehydrogenation and dry reforming pathways, respectively.

For the oxidative C-H and C-C bond cleavage on Pt-terminated-Ni₃Pt(111), the activation of *CH₃CH₂CH₃ is exothermic (ΔE = -0.49 eV) and has an E_a of 1.33 eV. Once *CH₃CH₂CH₂ is formed it can undergo dehydrogenation reaction to *CH₃CHCH₂ (ΔE = -0.75 eV; E_a = 1.07 eV) or react with *O to form *CH₃CH₂CH₂O (ΔE = -0.51 eV; E_a = 1.33 eV). Thus, *CH₃CH₂CH₂O formation is thermodynamically and kinetically more favorable than the C-H bond cleavage of *CH₃CH₂CH₂. That is, Pt-terminated-Ni₃Pt(111) promotes the C-C bond cleavage pathway (Figure 4b).

On bulk-terminated-Fe₃Ni(111), the activation of *CH₃CH₂CH₃ is more endothermic (ΔE = 0.43 eV) and with a higher activation barrier (E_a = 1.61 eV) than that of Ni₃Pt(111), in agreement with the experimental observation of lower propane conversion over the Fe₃Ni catalyst. Consistent with the experimental observation, the dehydrogenation of *CH₃CH₂CH₂ (ΔE = 0.29 eV; E_a = 1.02 eV) is thermodynamically and kinetically more favorable than the formation of *CH₃CH₂CH₂O (ΔE = 0.43 eV; E_a = 3.30 eV). Hence, in agreement with the predictions based on the values of ΔE and E_a, the Fe₃Ni(111) surface promotes the oxidative C-H bond cleavage pathway for the formation of propylene (Figure 4a). When Fe is oxidized and forms FeO on the surface, the activation of *CH₃CH₂CH₃ on Fe₃O₃/Ni(111) is still endothermic (ΔE = 0.63 eV) and has an E_a of 1.88 eV. Similar to Fe₃Ni(111), the dehydrogenation of *CH₃CH₂CH₂ (ΔE = -0.40 eV; E_a = 0.29 eV) is thermodynamically and kinetically more favorable than the formation of *CH₃CH₂CH₂O (ΔE = 0.01 eV; E_a = 2.13 eV). Hence, the FeO/Ni(111) interface promotes the oxidative C-H bond cleavage pathway for the formation of propylene (Figure 4c).

We used DFT calculated ΔE and E_a of selected reaction steps (Table below) to investigate the Brønsted–Evans–Polanyi (BEP) relationship. The trend in the values of ΔE and E_a are in general consistent with the BEP relationship. Overall, the predictions on dominant reaction pathway using ΔE are supported by that based on E_a. This table is now added as Table S9-5 in the SI.

Figure S9-4: (a) DFT calculated energy profiles for the oxidative C-H and C-C bond scission pathways on (b) $\text{Fe}_3\text{O}_3/\text{Ni}(111)$ and $\text{Fe}_6\text{O}_9/\text{Ni}(111)$.

Table S9-5: DFT calculated reaction energy (ΔE) and activation energy (E_a) (in eV) for selected reaction steps on Pt-ter- $\text{Ni}_3\text{Pt}(111)$, bulk-ter- $\text{Fe}_3\text{Ni}(111)$ and $\text{Fe}_3\text{O}_3/\text{Ni}(111)$.

Surface	$^*\text{CH}_3\text{CH}_2\text{CH}_3 + ^*\text{O} \rightarrow ^*\text{CH}_3\text{CH}_2\text{CH}_2 + ^*\text{OH}$		$^*\text{CH}_3\text{CH}_2\text{CH} + ^*\text{O} \rightarrow ^*\text{CH}_3\text{CH}_2\text{CH}_2\text{O} + ^*$		$^*\text{CH}_3\text{CH}_2\text{CH} + ^*\text{O} \rightarrow ^*\text{CH}_3\text{CHCH}_2 + ^*\text{OH}$	
	ΔE	E_a	ΔE	E_a	ΔE	E_a
Pt-ter- $\text{Ni}_3\text{Pt}(111)$	-0.49	1.33	-0.75	1.07	-0.51	1.33
Bulk-ter- $\text{Fe}_3\text{Ni}(111)$	0.43	1.61	0.43	3.30	0.29	1.02
$\text{Fe}_3\text{O}_3/\text{Ni}(111)$	0.63	1.88	0.01	2.13	-0.40	0.29

[Action]

The following discussion has been added in the revised manuscript:

Starting on line 173:

“Overall the DFT results reveal that the C-C bond cleavage pathway is preferred on $\text{Ni}_3\text{Pt}(111)$, while bulk-terminated- $\text{Fe}_3\text{Ni}(111)$ favors the C-H bond cleavage pathway. Kinetically, this is also the case based on the comparison of activation energies. According to the DFT calculations, on Pt-terminated- $\text{Ni}_3\text{Pt}(111)$, the $^*\text{O}$ insertion reaction ($^*\text{CH}_3\text{CH}_2\text{CH}_2 + ^*\text{O} \rightarrow ^*\text{CH}_3\text{CH}_2\text{CH}_2\text{O} + ^*$) along the C-C bond cleavage pathway ($\Delta E = -0.75$ eV and $E_a = 1.07$ eV) is thermodynamically and kinetically more favorable than the oxidative dehydrogenation reaction ($^*\text{CH}_3\text{CH}_2\text{CH}_2 + ^*\text{O} \rightarrow ^*\text{CH}_3\text{CHCH}_2 + ^*\text{OH}$) along the C-H bond cleavage pathway ($\Delta E = -0.51$ eV and $E_a = 1.33$ eV). In contrast, on bulk-terminated- $\text{Fe}_3\text{Ni}(111)$, the oxidative dehydrogenation reaction ($\Delta E = 0.29$ eV and $E_a = 1.02$ eV) is more favorable than the $^*\text{O}$ insertion reaction ($\Delta E = 0.43$ eV and $E_a = 3.30$ eV).

Starting on line 198:

“Again, such thermodynamic predictions are fully supported by the calculated E_a , showing that the oxidative dehydrogenation reaction ($\Delta E = -0.40$ eV and $E_a = 0.29$

eV) is more favorable than the *O insertion reaction ($\Delta E = 0.01$ eV and $E_a = 2.13$ eV) on the $\text{Fe}_3\text{O}_3/\text{Ni}(111)$ surface.”

The following sentences have been added to the SI section S9

“The transition state of a chemical reaction was located using the climbing image nudged elastic band (CI-NEB) method implemented in VASP.³ The activation energy (E_a) of a chemical reaction is defined as the energy difference between the initial and transition states while the reaction energy (ΔE) is defined as the energy difference between the initial and final states.”

There are also other comments to be made on the DFT work:

(2) The authors derive the model surfaces from XANES data, which is a very good starting point, indicating reduced Ni and oxidized Fe. However, the 2 catalyst models $\text{Fe}_3\text{Ni}(111)$ and $\text{FeO}/\text{Ni}(111)$ are limiting cases, and the actual catalyst surface may lay in between. Nothing is mentioned on the matter, and the surface energies of both states are not compared. Furthermore, will (111) still be the dominating facet in the case of an FeO top layer? And are the results sensitive to the geometry/size of the FeO ‘top layer’, since this is currently modelled as a 6-Fe-atom cluster on top of Ni(111)?

[Response]

From the response to Reviewer 1, Comment 8, “Two models were used to describe the Fe_3Ni cluster supported on CeO_2 . One was the bulk terminated $\text{Fe}_3\text{Ni}(111)$ surface using the L12 cubic crystal structure and a four layer 4×4 surface slab (Figure S9-1a). The other was the FeO_x clusters supported on Ni(111), where both Fe_6O_9 and Fe_3O_3 clusters on 3 layer 7×7 Ni(111) and 5×5 Ni(111) surfaces (Figure S9-1c) were considered. In this way, the experimentally observed oxidation of Fe on the surface under reactions can be captured. Similarly, the Ni_3Pt cluster supported on CeO_2 was described by $\text{Ni}_3\text{Pt}(111)$ surface using the L12 cubic crystal structure and a four layer 4×4 surface slab (Figure S9-1b), where the formation of Pt skin on the surface was included to consider the segregation of Pt according to our experimental observations.^[2] In both cases, the effect of CeO_2 and the particle size was not taken into consideration. According to our experimental results, CeO_2 plays a crucial role in the initial CO_2 activation, while the subsequent reforming and dehydrogenation reactions take place on the surface of bimetallic particles. The different trend in the catalytic behavior between $\text{Fe}_3\text{Ni}/\text{CeO}_2$ and $\text{Ni}_3\text{Pt}/\text{CeO}_2$ observed experimentally during propane oxidative dehydrogenation is mostly associated with the surface of bimetallic alloys, which justifies our focus on the trend of different bimetallic surfaces in the DFT calculations.”

We agree with the Reviewer that the binding energies of intermediates vary with the size of the FeO clusters on Ni(111). However, the overall trend of selectivity may not change. To test this hypothesis, we computed energy profiles for the C-H and C-C bond cleavage using a smaller $\text{Fe}_3\text{O}_3/\text{Ni}(111)$ model. It is found that, trends observed using $\text{Fe}_3\text{O}_3/\text{Ni}(111)$ are not significantly

[3] G. Henkelman, B. P. Uberuaga, H. Jónsson, G. Henkelman, *J. Chem. Phys.* **2000**, *113*, 9901–9904.

different from those obtained using Fe₆O₉/Ni(111) (Figure S9-4). Since these calculations were computationally demanding, we were unable to test this hypothesis using a FeO cluster larger than Fe₆O₉.

[Action]

The following modification has been made to the revised description in the SI section S9 and Figure S9-4 is also now available.

“The binding energies of intermediates vary with the size of the FeO clusters on Ni(111). However, the overall trend of selectivity may not change. To test this hypothesis, we computed energy profiles for the C-H and C-C bond cleavage using a smaller Fe₃O₃/Ni(111) model. It is found that, trends observed using Fe₃O₃/Ni(111) are not significantly different from those obtained using Fe₆O₉/Ni(111) (Figure S9-4). Since these calculations were computationally demanding, we were unable to test this hypothesis using a FeO cluster larger than Fe₆O₉.”

(3)- There is no discussion at all on how the adsorption sites to which the intermediates bind have been identified, though there are many different possible sites, particularly for the Fe₃Ni and FeO/Ni models. This should at least be briefly discussed in the Supporting Information: default geometry optimization will not automatically evaluate all possible sites. The reader can currently not know that the identified states really are the minimum energy states.

[Response]

We thank the Reviewer for this valuable comment. We have tested various possible adsorption sites during our geometry optimization calculations.

[Action]

The tested adsorption sites are marked in Figures S9-(1-3) and the corresponding binding energies are listed in Tables S9-1, S9-2 and S9-3 in the SI section S9.

(4a) From figure 4 is derived that Fe₃Ni is more selective for propylene, while Ni₃Pt is more active and CO-selective. This is indeed a likely possibility based on the data shown. However, the discussion of Figure 4 is more positive than the Figure actually looks like. Firstly, the schemes go up to 3 eV and down to -4 eV, which is enormous (and barriers are not even included, which will make them even higher in energy). Most likely, both schemes lead to complete inactivity because of too high barriers (for a and c) or poisoning by ethyl, CO and/or O species (for b). Secondly, the first step for (a) is propane adsorption/activation with a DeltaE of +0.8 to +1.2 eV, depending on the path. In terms of DeltaG this will be even higher by about 0.9 eV at 873 K, which will lead to negligibly low coverages.

(4b) And finally, because the final gas phase states are not included in the plot (with gas phase O, CO, ...) there is no final check on the correctness of the DFT data. Trying to interpret the current data, one can only come up that in case of (a) and (c) energy will be released upon the desorption

of ethyl, CO and O - which is unlikely – or these species will never desorb at all in case of (b), depending where the final gas phase states are.

[Response 4a]

The energy diagrams in Figure 4 show that propane activation and the subsequent steps are uphill in energy significantly. This suggests that activation and conversion of propane could only be achieved at high temperatures. This is consistent with our experimental observation during activation barrier measurements, which indicate that temperatures greater than 800 K are required for quantifiable propane conversion rates.

We agree with the Reviewer that the activation of propane could be a difficult step and thus controls the activity/conversion on the PtNi and FeNi bimetallic catalysts. According to the DFT results in Figure 4, the activation of propane is more difficult on Fe₃Ni(111) and FeO/Ni(111) than on Pt-ter-Ni₃Pt(111). This prediction is further corroborated by the DFT calculated activation energies (E_a) of the oxidative propane activation ($*\text{CH}_3\text{CH}_2\text{CH}_3 + *\text{O} \rightarrow *\text{CH}_3\text{CH}_2\text{CH}_2 + *\text{OH}$) on Pt-terminated-Ni₃Pt(111) and bulk-terminated-Fe₃Ni(111) surfaces and at the FeO/Ni(111) interface using a smaller Fe₃O₃/Ni(111) model. It is found that the E_a for the oxidative propane activation is 1.33 eV, 1.61 eV and 1.88 eV on Pt-terminated-Ni₃Pt(111), bulk-terminated-Fe₃Ni(111) and Fe₃O₃/Ni(111), respectively. Consistent with our experimental observations in Table 1, DFT predicts lower conversion on FeNi than PtNi catalysts. Therefore, the DFT results obtained from the current models qualitatively describe the trends in conversion observed on the FeNi and PtNi catalysts.

[Response 4b]

The DFT calculations were used to estimate the change in Gibbs free energy (ΔG) for the reaction $\text{CH}_3\text{CH}_2\text{CH}_3(\text{g}) + \text{CO}_2(\text{g}) + * \rightarrow \text{CH}_3\text{CHCH}_2(\text{g}) + \text{H}_2\text{O}(\text{g}) + \text{CO}(\text{g}) + *$ at $T = 298.15 \text{ K}$. The DFT calculated $\Delta G = 1.55 \text{ eV}$ is close to the experimental value of 1.20 eV .⁴

CO is one of the common products in both the C-C and C-H bond cleavage pathways. The DFT calculations show that the binding energies of CO are -1.30 eV, -1.92 eV and -2.04 eV on Pt-terminated-Ni₃Pt(111), bulk-terminated-Fe₃Ni(111) and FeO/Ni(111) interface, respectively. Since the reaction occurs at high temperature (823 K), the desorption of CO from the three surfaces is expected to be a facile process due to the contribution of entropy at such high temperature. Ethyl (C₂H₅) is one of the reaction intermediates that undergoes O-insertion, C-H and C-C bond scission reactions to eventually produce CO, and H₂. According to the DFT calculations, the binding energies of O are -3.29 eV, -6.23 eV and -5.78 eV on Pt-terminated-Ni₃Pt(111), bulk-terminated-Fe₃Ni(111) and FeO/Ni(111) interface, respectively. As a result, *O on Pt-terminated-Ni₃Pt(111) reacts with C_xH_y species to form *C_xH_yO intermediate, which promotes the C-C bond scission. In contrast, the more stable *O on bulk-terminated-Fe₃Ni(111) and the FeO/Ni(111) interface is expected to remain on the surface. Such adsorbed *O then facilitates the selective C-H bond scission of propane to produce propylene.

[4] J. A. Dean, in *Lange's Handb. Chem.*, McGraw-Hill, New York, 1985.

[Action 4a]

Please see the action for Comment 1 of Reviewer #2 where the appropriate revisions have been made to the manuscript in lines 173 and 198.

[Action 4b]

The following sentences have been added to section S9 of the SI:

“The current setup in the DFT calculations predicted the Gibbs free energy (ΔG) as 1.55 eV for the reaction $\text{CH}_3\text{CH}_2\text{CH}_3(\text{g}) + \text{CO}_2(\text{g}) + * \rightarrow \text{CH}_3\text{CHCH}_2(\text{g}) + \text{H}_2\text{O}(\text{g}) + \text{CO}(\text{g}) + *$ at $T = 298.15 \text{ K}$, which is close to the experimental value of 1.20 eV.^{[7]”}

The following sentences have been added to the manuscript on line 203:

“Finally, on the three surfaces studied the desorption of $*\text{CO}$ is expected to be a facile process due to the contribution of entropy at 823 K. $*\text{C}_2\text{H}_5$ is one of the reaction intermediates that undergoes O-insertion, C-H and C-C bond scission reactions to eventually produce CO , and H_2 . The $*\text{O}$ species on Pt-terminated- $\text{Ni}_3\text{Pt}(111)$ react with $*\text{C}_x\text{H}_y$ to form the $*\text{C}_x\text{H}_y\text{O}$ intermediate, which promotes the C-C bond scission. In contrast, the more stable $*\text{O}$ on bulk-terminated- $\text{Fe}_3\text{Ni}(111)$ and the $\text{FeO}/\text{Ni}(111)$ interface are expected to remain on the surface, which facilitates the selective C-H bond scission of propane to produce propylene.”

(5) As a minor comment, it is confusing that the cited selectivities in the text (e.g. page 4, line 75) are not the same as those in Table 1. I realize the difference (250 minutes on stream vs. averaged out between 10 to 12 hours on stream), but it is confusing.

[Response]

We thank the Reviewer for the comment that the statement on line 75 can cause confusion.

[Action]

The reported value has been updated to reflect the steady state amount on line 78, which is the average of hours 10-12 on stream as specified in the caption of Table 1.

Summarized, I endorse publication of this manuscript, but would suggest at least a more careful representation and interpretation of the DFT data, and preferably also suggest the calculation of reaction barriers – at least of the steps that are considered decisive for the determination of the selectivities.

Reviewer #3:

The manuscript by Gomez et. al proposes the use of Ni-based bimetallic catalysts for CO₂-based ODH or propane dry reforming processes to produce propylene or syngas while consuming CO₂. The use of CO₂ instead of O₂ as an oxidant is described to have multiple benefits: (1) The consumption of a greenhouse gas that is typically a waste stream, (2) a shift of dehydrogenation reaction thermodynamics to allow for higher conversion of the alkane at lower reaction temperatures than non-oxidative dehydrogenation or conventional dry reforming operations, and (3) the milder oxidizing power of CO₂ may reduce unselective reactions. The authors present NiFe/CeO₂ as a primarily dehydrogenation catalyst and NiPt/CeO₂ as a reforming catalyst, indicating that the close interaction of the two metals plays a role in determining process selectivity.

The most important contribution of this study was the attempt to modify nickel-based systems via bimetallic interactions to tune reaction selectivity to either dehydrogenation or reforming processes. This approach would provide valuable insight on how different metal interactions affect reactivity to the broader catalysis community. Beyond the product selectivity, the bimetallic interactions may change catalyst stability, which is also an important factor for future development. This contribution, however, shows several shortcomings that limit its impact and need to be addressed for a more complete study.

[Response]

We thank the Reviewer for the thorough evaluation of our work. We have made careful revisions according to the Reviewer's comments. Please see below for the individual concerns.

(1a) First, the prepared catalysts appear to systematically replace one metal with another to explore their effect, but the FeNi and NiPt catalysts are different not only in their metal composition but also their atomic ratios. The authors should maintain the Ni:Metal ratio constant, which is not the case in the current work as we see Fe:Ni= 3:1 and Pt:Ni = 1:3.

(1b) This is particularly important in the Fe₃Ni₁ sample, which is shown by EXAFS to possess Fe-O-Fe bonds. This result, however, may be in part due to the larger Fe motifs caused by higher Fe loadings compared to Ni. If Fe dopes a Ni particle (in contrast to Ni doping an Fe particle) would the EXAFS still show a similar structure? Without similar compositions, it becomes complicated to compare bimetallic catalysts, especially at the low loadings used in the manuscript.

[Response 1a]

The approach utilized to determine metal loading is one that has been extensively utilized within our research group and has been maintained to build a library of catalysts for different classes of catalytic reactions. The reference loading of precious metal within the group is 1.67 wt.% Pt. For bimetallic synthesis, Pt is replaced with another metal with an equivalent atomic concentration. In the Fe₃Ni₁/CeO₂ catalyst, Ni₁ replaces Pt and then the Fe loading is calculated such that for every atom of Ni there are 3 atoms of Fe.

The authors agree that a comparison among Fe₃Ni and Ni₃Pt to Ni₃Fe is of great interest and the flow reactor results are provided below in a modified Table 1 that is now available in the SI section S7-3. Changing the atomic ratio from Fe₃Ni to Ni₃Fe increases the activity but

completely alters the selectivity to favor the dry reforming of propane, this is true even at comparable C₃H₈ reactant conversion.

[Response 1b]

Furthermore, to investigate the effect of higher Ni content on the oxidation state of a Ni-Fe catalyst in-situ XANES and EXAFS measurements were conducted. It can be seen from Figure R3-1 that the Ni₃Fe catalyst also contains oxidized iron and metallic Ni, like the Fe₃Ni counterpart. Table R3-1 lists the bond distances and coordination numbers (CN) per examined shell on both the Ni and Fe K edges. The very similar CN values of the two catalysts suggest that Ni₃Fe and Fe₃Ni have a similar bulk physical structure. From our DFT calculations, we understand that on the clean Fe₃Ni(111) surface Ni segregation is thermodynamically favorable, but in the presence of oxygen, Fe segregates to the surface due to the stronger Fe-O bond (please see the response to Reviewer #2 Comment 9). On a surface that contains a greater amount of Ni it is reasonable to believe that Ni segregation will be dominant. This may essentially dampen the catalytic effect observed with oxidized Fe and therefore promote activity like that of monometallic Ni, which is toward the dry reforming reaction. This indeed correlates with the experimental flow reactor results which show Ni₃Fe to favor dry reforming and Fe₃Ni to favor propylene production.

[Action 1a & 1b]

The flow reactor results of Ni₃Fe have been added to the SI section S7-3 with the following discussion,

“Furthermore, to compare the activity of Ni₃Pt with a catalyst containing an equal amount of Ni, Ni₃Fe was also evaluated. When the atomic ratio of the Fe₃Ni catalyst is changed to Ni₃Fe, the catalytic activity becomes higher, leading to 26.9% CO₂ and 7.4% C₃H₈ conversion. However, the reaction pathway favors DRP even at comparable propane conversion (Modified Table 1) with a CO selectivity of 90.6%. Thus, when Ni is coupled with non-precious Fe at a ratio of 1:3, higher dehydrogenation selectivity is achieved, and propylene is produced. In contrast, when Ni is alloyed with precious metal Pt, the reforming activity is enhanced compared to Ni₃Fe (Modified Table 1) and monometallic Ni₃ (Figure S7-3a).”

The reader is directed to section S7-3 by the following sentence in the revised manuscript,

“Further analysis, such as the comparison of CeO₂ supported Ni₃Pt with Ni₃Fe and Fe₃Ni catalysts along with CO selectivity following CO₂ conversion plots can be found in the SI sections S7-3 and S2-3b, respectively.”

Modified Table 1. Catalyst flow reactor results for CO₂ + C₃H₈ reaction (10 mL/min each) at 823 K with Ar diluent (20 mL/min) and 100 mg of catalyst (16-20 mesh). Catalysts marked with an asterisk indicate that the sample was diluted to achieve comparable C₃H₈ reactant conversion to Fe₃Ni. Values are obtained by averaging data from hours 10-12. Selectivity and yield are on a C₃H₈ basis (including only carbonaceous species). Catalysts are synthesized by atomic ratios corresponding to a 1.67 wt.% Pt₁ basis, thus the weight percent of Fe₃, Ni₁, and Ni₃ are 1.43, 0.5, and 1.5, respectively. The nomenclature assigned by subscripts such as in Fe₃Ni means that there are 3 atoms of Fe for every atom of Ni.

Catalyst /CeO ₂	CO uptake μmol·g ⁻¹	Conversion (%)		TOF (site ⁻¹ ·min ⁻¹)		Selectivity (%)					Yield (%)	
		CO ₂	C ₃ H ₈	CO ₂	C ₃ H ₈	CO	C ₃ H ₆	CH ₄	C ₂ H ₆	C ₂ H ₄	CO	C ₃ H ₆
Fe ₃ Ni	31.9	4.0	2.7	5.7	3.4	40.2	58.2	0.8	0	0.8	1.1	1.6
Fe ₃ Pt	31.5	2.6	1.1	3.5	1.5	65.1	32.0	1.3	0	1.6	0.7	0.3
Ni ₃ Pt	50.1	39.4	11.6	37.5	10.5	96.2	2.8	0.83	0.1	0	11.1	0.3
Ni ₃ Fe	36.0	26.9	7.4	34.5	9.2	96.4	2.9	0.63	0.03	0	7.2	0.2
*Ni ₃ Pt	-	7.8	2.2	-	-	87.8	11	0.9	0.0	0.3	2.0	0.2
*Ni ₃ Fe	-	7.9	2.1	-	-	90.6	8.2	0.8	0.0	0.4	1.9	0.2
Ni ₁	13.1	9.3	3.0	31.9	8.9	86.8	12.3	0.60	0.24	0	2.6	0.4
Ni ₃	37.7	32.8	9.6	40.2	11.4	94.9	2.9	2.11	0.05	0.06	9.1	0.3
Pt ₁	16.0	4.2	1.6	8.1	2.8	77.0	21.2	0.8	0.9	0	1.3	0.4

(2) Comparison of the catalyst via catalytic testing present NiFe as a dehydrogenation catalyst while NiPt is a reforming catalyst. Yet it is difficult to compare both Ni-based catalysts in this way (as seen in Table 1) when the product selectivities are not shown at iso-conversion. Judging by the fact that the NiFe catalyst has a C₃H₆ selectivity of about 60% at 2.7% C₃H₈ conversion, it is reasonable to expect this selectivity to decrease when testing this catalyst at the same conversion shown for NiPt. While I understand it is difficult to compare catalysts with different rates of deactivation, such an important claim of the work clearly requires a better comparison set. Any claims of product distribution differences must be done at isoconversion to be valid.

[Response]

The authors agree that selectivity for different catalysts should be examined at comparable reactant conversion. Results for flow reactor studies conducted to achieve comparable C₃H₈ conversion are available in the SI section S7. The reader is referred to section S7 in the manuscript on lines 85-86, "... a robust selectivity toward CO of 88% at comparable reactant conversions (Table S7-1) ..."

[Action] To avoid misinterpretation and facilitate proper selectivity comparison, the flow reactor results conducted to achieve comparable propane conversion are moved from SI to Table 1 of the manuscript and the following sentence has been added to the caption of Table 1,

"Catalysts marked with an asterisk indicate that the sample was diluted to achieve comparable C₃H₈ conversion to Fe₃Ni."

(3a) Furthermore, the authors should show conversion-selectivity trends to illustrate the differences between the catalysts and determine primary reaction products from their selectivity extrapolated to zero conversion (i.e. the y-intercept of selectivity in a conversion-selectivity plot). These selectivity trends would further validate their claims of CO being produced from reforming activity rather than simply overoxidation processes that CO₂ may ameliorate. I expect the authors must have this data readily available. All these considerations seek to answer unequivocally what are the main reaction pathways involved with these bimetallic materials.

(3b) The NiFe catalyst in particular, has what amounts to almost 60-40 distribution of C₃H₆ and CO selectivities, which is hardly comparable to Cr-based catalysts for CO₂ – ODHP which show selectivities well above 80% towards propylene. I would be more compelled to agree with their claims if they presented more evidence to show that the produced CO does not come from reforming activity.

[Response 3a]

Based on the Reviewer's comment CO₂ conversion vs CO selectivity plots for both bimetallics are depicted below in Figure S2-3b-1. For Ni₃Pt, the y-intercept of CO selectivity is 97%, suggesting it is a typical reforming catalyst. While for Fe₃Ni, the y intercept of CO selectivity is 22.2%. The primary reactions of CO₂ + C₃H₈ include the dry reforming [Equation 1] and CO₂-ODHP [Equation 2], which can occur simultaneously at temperatures above 823 K.

Other relevant reactions for propane dry reforming include propane decomposition to carbon, H₂, and CH₄ [Equation 3], the reverse Boudouard reaction [Equation 4], the reverse water-gas shift [Equation 5], and direct propane dehydrogenation [Equation 6] as listed below.

The amount of CO produced from C₃H₈ can be calculated via the oxygen balance (i.e., all the oxygen in CO and H₂O comes from CO₂),

$$F_{\text{CO originated from CO}_2}^{\text{outlet}} = \frac{F_{\text{CO}}^{\text{outlet}} + F_{\text{H}_2\text{O}}^{\text{outlet}}}{2} \quad (1)$$

$$F_{\text{CO originated from C}_3\text{H}_8}^{\text{outlet}} = F_{\text{CO}}^{\text{outlet}} - F_{\text{CO originated from CO}_2}^{\text{outlet}} \quad (2)$$

where *F* is the flow rate of reactant or product in mol/min. If no over-oxidation were to occur in the above reaction scheme, then the total CO produced would be equivalent to the sum of C₃H₆ and H₂O produced. However, over the Fe₃Ni catalyst there is more CO than the sum, (Figure S2-3b-2), indicating that CO can also be produced via over-oxidation of olefins/lighter components

and/or the reverse Boudouard reaction. It can be assumed that over-oxidation is included in reforming because all the olefins, lighter components, and C(s) are produced from C₃H₈ and are then oxidized by CO₂ to finally produce CO. Among one of the fundamental differences between CO₂-ODHP and conventional oxidative dehydrogenation is the net reduction of CO₂ and the production of CO from over-oxidation, while the latter reaction only produces more CO₂ upon over-oxidation.

[Response 3b]

As pointed out by the Review, Cr-based catalysts have been reported in the literature for CO₂ oxidative dehydrogenation. In general, chromium catalysts require high temperatures, are subject to coking, and upon reaching a maximum chromium loading alkane activity cannot be increased further. While chromium-based catalysts show high initial conversion and selectivity, the implementation is limited due to short lifecycles of the catalysts and high toxicity of chromium. Thus, special safety considerations must be taken throughout the lifetime of the catalyst (preparation, use, and disposal of). There is a strong interest in developing safer and more environmentally benign catalysts that are active at low temperatures for the CO₂-ODH of alkanes. This prompted our group to explore bimetallic catalysts Fe₃Ni and Ni₃Pt for the reactions of CO₂+C₃H₈.

[Action 3a & 3b]

The response above for Comment 3a has been added to the SI section S2-3b. The CO selectivity vs CO₂ conversion plots for both catalysts along with the total CO production for Fe₃Ni have also been added to section S2-3b. The following sentence has been added to line 89 of the manuscript,

“Further analysis, such as the comparison of CeO₂ supported Ni₃Pt with Ni₃Fe and Fe₃Ni catalysts along with CO selectivity following CO₂ conversion plots can be found in the SI sections S7-3 and S2-3b, respectively.”

Figure S2-3b-1: CO₂ conversion vs CO selectivity plots for Fe₃Ni and Ni₃Pt bimetallics supported on CeO₂.

Figure S2-3b-2: Total CO production and total CO production subtracting contributions from CO₂-ODHP and RWGS following time on stream.

(4) The kinetic experiments carried out (reaction order and apparent activation energy), while properly carried out, seem redundant or not used to complement the rest of their presented work. The authors highlight the competitive adsorption of C₃H₈ and CO₂ by comparing the effect of one substrate on the rate of consumption of the other (i.e. the effect of P_{CO₂} on rate_{C₃H₈}) but they do not show the effect of P_{C₃H₈} on rate_{C₃H₈}, for instance. In conventional ODHP with oxygen, there is a first order dependence on P_{C₃H₈} due to the weak adsorption of the alkane on the active site, however I would not expect this to be the case if the adsorption-like dependence shown in Figure S2-2 holds. Complementing this figure with the other reaction order experiments would present a more complete picture with which to develop an overall rate equation, for instance.

[Response]

On the Fe₃Ni catalyst, as the CO₂ partial pressure (P_{CO₂}) increases the rate of CO₂ consumption increases linearly ($y = 0.25x - 10.5$, $R^2 = 0.996$). Meanwhile, the propane consumption rate is initially unaffected and then starts to decline at higher P_{CO₂}. As the C₃H₈ partial pressure (P_{C₃}) increases over the Fe₃Ni catalyst, the rate of C₃H₈ consumption initially increases until reaching a C₃H₈: CO₂ ratio of 1:1, after which the rate begins to decline, as shown in Figure R3-4c. Thus, high P_{C₃} hinders the reaction since both the consumption rate of CO₂ and C₃H₈ reach a maximum at a C₃H₈: CO₂ ratio of 1:1. Over the Ni₃Pt catalyst, as the CO₂ partial pressure increases the rate of CO₂ consumption increases linearly ($y = 0.51x - 8.9$, $R^2 = 0.980$). While the CO₂ consumption rate is not affected by increasing the P_{C₃} over Ni₃Pt (Figure R3-5c), the propane consumption rate appears to be initially positively affected (Figure R3-5d) but then the rate also declines. Qualitatively, these observations are consistent with a competitive adsorption mechanism at high propane partial pressure over both catalysts.

[Action]

The partial pressure dependences for both reactants are provided in the SI as Figures S2-2 and S2-3. A sentence has been added to the manuscript that summarizes the main conclusion of the experiments that explore the effect of reactant_A/reactant_B partial pressure on the rate of reactant_A/reactant_B on line 111:

“Particularly, the rates for both reactants decrease at high propane partial pressure, suggesting that as the reaction progresses intermediates from propane block surface sites and lead to a loss in activity.”

Figure S2-2: Effect of CO₂ partial pressure on the C₃H₈ and CO₂ consumption rate (a & b) and the effect of C₃H₈ partial pressure on CO₂ and C₃H₈ consumption rate (c & d) over Fe₃Ni.

Figure S2-3: Effect of CO₂ partial pressure on the C₃H₈ and CO₂ consumption rate (a & b) and the effect of C₃H₈ partial pressure on CO₂ and C₃H₈ consumption rate (c & d) over Ni₃Pt.

(5) Similarly, the activation energy experiments only present the activation energy of the purportedly primary reaction pathway on each catalyst (Figure S2-1). To complement these, the activation energy of the minor pathway (i.e. reforming with the NiFe catalyst) should be presented. If the authors' hypotheses are valid, the activation energies should mirror the results of their DFT calculations however, as discussed below, the DFT calculations cannot be directly compared due to the lack of energy barriers calculated.

[Response]

Based on the Reviewer's comment activation barrier values are presented below for reactant consumption rate of C₃H₈ and CO₂ as well as the product formation rate of C₃H₆ and CO. Ni₃Pt exhibits less than 3% selectivity toward propylene, therefore the activation barrier over C₃H₆ cannot be accurately obtained over the temperature range evaluated. The comparison with DFT calculated barriers will be addressed in the next comment.

	Activation Barrier Values (kJ/mol)			
	C ₃ H ₈	CO ₂	C ₃ H ₆	CO
Fe ₃ Ni /CeO ₂	110	135	115	121
Ni ₃ Pt/CeO ₂	109	123	-	119

Table S2-1: Activation barrier values for Fe₃Ni and Ni₃Pt.

[Action]

The values from the above table are now included in the SI section S2-4.

(6) Related to the experimental reactivity results, the authors present DFT calculations that suggest that the NiFe and FeO-Ni surfaces show favorable energy pathways for dehydrogenation. These claims, however, are based on adsorption enthalpies, which represent only part of the overall picture of a chemical reaction. Any claim to compare both C-H and C-C bond breaking requires activation energy calculations, which would also complement the apparent energy barriers observed in experiment and inform of the kinetics of these two reactions. It is unclear to me why energy barriers would not be included in such study.

[Response]

The DFT calculations of the selected C-H and C-C bond scissions are discussed in the Response to Comment (1) from Reviewer #2, as these same concerns were raised. Due to the very large numbers of elementary reaction steps, only selected activation barriers were calculated to highlight the general trend in the activation of the C-H and C-C bonds. Because we were unable to calculate the activation energies for the entire reaction network due to computation limitations, it would not be possible to provide a meaningful comparison of the activation barriers between DFT and experiments. We appreciate the importance of the comment by the reviewer, and would consider further DFT calculations in future studies.

(7) Furthermore, the DFT calculations show that the NiFe and FeO-Ni surface could favor C-H bond breaking but the authors never discuss why there is such a significant amount of CO still being formed with the NiFe catalyst, which I would presume is due to the unselective overoxidation of the olefin products.

[Response]

The Reviewer is correct. As provided in the Response to Comment 1 of Reviewer #2, the activation barrier for the oxidative dehydrogenation (C-H bond scission) and *O insertion reaction (responsible for subsequent C-C bond cleavage) is 1.02 eV and 3.30 eV, respectively, on Fe₃Ni/Ni(111). Similarly, the activation barrier for these two steps is 0.29 eV and 2.13 eV, respectively, on Fe₃O₃/Ni(111), confirming that both surfaces favor the C-H bond scission. Regarding the experimental observation of the significant amount of CO formation, as described in the Response to Comment 3 for the current Reviewer (Figure S2-3b-2), over-oxidation of olefins/lighter components and/or the reverse Boudouard reaction also contribute to the total amount of CO production.

[Action]

As described in the Response to Comment 1 from Reviewer #2, the related discussion and activation barriers have been added to the revised manuscript.

Other minor issues to address:

(8) The TGA analysis presented in Figure S5-1 aims to show that the NiFe catalyst does not coke significantly, but the comparison is not exactly fair as the two catalysts were run at significantly different conversions. The catalysts should be run at comparable levels of conversion and then their mass change compared. While the result may be the same as presented, the experimental approach needs to be improved. Furthermore, it is unclear why the figure itself is presented with two temperature axes increasing in opposite directions of the x-axis. This approach is confusing and may lead to misinterpretation. If the authors wish to prevent overlap of the profiles, they should use two figures instead.

[Response]

We thank the Reviewer for the comment and agree that the temperature axis on the DTGA plot can lead to misinterpretation. To address the concern regarding a fair comparison, the diluted Ni₃Pt spent sample was subjected to TGA measurement. The diluted Ni₃Pt sample, which has comparable propane conversion to Fe₃Ni, also lost less than half a percent and does not have a significant DTGA peak. Thus, the coking over the Ni₃Pt catalyst at comparable propane conversions to Fe₃Ni is not significant.

[Action]

The DTGA vs temperature plot in the SI section S5 has been modified to include the diluted Ni₃Pt sample and contains only one temperature axis. In addition, the following sentence has been added to the manuscript on line 129,

“However, the coking over the Ni₃Pt catalyst at comparable propane conversion to Fe₃Ni is not significant.”

Figure S5-1: DTGA results for Fe₃Ni, Ni₃Pt, and diluted Ni₃Pt supported on CeO₂.

(9) The EXAFS fitting shown in Figure S10-1 seems incomplete, as the Ni fit is only reasonable at the 2Å distance while the rest of the experimental data does not agree with the fit. I would expect proper fitting up to at least 3Å for any valid bond distance claims to be made. The Fe edge is somewhat better but still should be improved. If the authors explained how they approached their fitting procedure, the reader may be able to understand why the fits shown are deemed valid when intuitively they seem incomplete.

[Response]

Collecting the fluorescence signal of 3d metals on a support composed by high Z element (CeO₂ in our system) usually results in relatively weak signals, due to the strong absorption of the oxide support as well as the relatively strong absorption of window materials at low energy edges. In our catalysts, the Fe and Ce fluorescence signals strongly overlap, making the Fe signal very weak. The energy range of Ni was also affected by the Cu signal (our in-situ microchannel cell is made by Cu). In addition, the data were all collected at 873 K and at such high temperature the high thermal displacement would dump the high k data more strongly. Therefore, the data shown in Figure S10-1 were noisy above $k = 10 \text{ \AA}^{-1}$ (see the Figure S10-B), which would prevent us from obtaining good fitting for the relatively weak features between the distance of 2Å-3Å.

Figure S10-B: k space for Fe and Ni for the fitting shown in Figure S10-1.

Furthermore, the independent points are not sufficient. In our fitting, both Ni and Fe spectra only have about 10 independent points, which could support at most 3-shell fitting (k range from 2.7 to 10.2 with an R range from 1.2 to 3.2~3.6). To reduce the correlation between parameters, constraints and restraints are required. However, since a heterogeneous catalyst is a mixture rather than a compound, there are no good constraints to make. As a result, we make minimum structural hypothesis based on the XANES in order not to introduce subjective mistakes. For Ni, metallic Ni foil is used, while for Fe, Fe₂O₃ and Fe₃O₄ are the models we used. We plan to design in-situ cells (without using Cu) and collect XAS data on more high flux beamline to obtain better fittings in our future synchrotron measurements.

[Action]

EXAFS fitting procedure as described above has been added to the SI section S10.

(10a) Based on the incomplete reactivity studies, different catalyst compositions and lack of energy barriers/transition states in the DFT portion of the work I cannot recommend this manuscript for publication in a high-impact publication such as Nature Communications.

(10b) Furthermore, the catalytic performance that is presented shows dehydrogenation activity far below the currently state of the art and no inherent benefit of these materials seems to be proposed. The authors should significantly modify their experimental approach to improve this work to be publication-ready in another more specialized journal.

[Response 10a]

We thank the Reviewer for the careful consideration of our work and hope that the extensive revisions, including the significant amount of additional DFT calculations and reactor studies have sufficiently addressed the very useful comments by the Reviewer. Flow reactor results conducted at comparable propane conversion are now included in Table 1 for easier accessibility. The catalytic performance of Fe₃Ni/CeO₂ and Ni₃Pt/CeO₂ have been compared to counterpart Ni₃Fe/CeO₂, which shows preference toward reforming. Expanded kinetic studies that evaluate the effect of reactant_{A/B} partial pressure on reactant_{A/B} consumption rate strengthen the conclusion of competitive adsorption and indicate that high propane partial pressures hinder the reaction rates of both C₃H₈ and CO₂. Activation barrier values for each reactant and main products have been added. We have also stretched our computation limit in the past three months to provide additional DFT calculations of activation barriers for selected reactions to provide a general trend in the C-H and C-C bond scissions over the different catalysts. We believe that the content of this work is substantially improved and is now suitable for Nature Communications.

[Response 10b]

While the activity of the Fe₃Ni/CeO₂ catalyst for propylene production is low compared to certain Cr-based catalysts, the non-toxic nature of the Fe₃Ni bimetallic catalyst has several benefits such as activity at lower temperatures and lower coke formation. Cr-based catalysts require meticulous care throughout their short lifecycles and a more benign catalyst is needed for large scale operations, especially for those that can utilize CO₂. Additionally, propylene production is experiencing a huge gap as the demand exceeds production. Current on-purpose propylene methods are highly endothermic and release large amounts of CO₂. CO₂-ODHP can help bridge the propylene gap while achieving lower operating temperatures while simultaneously mitigating CO₂ emission.

Reviewers' Comments:

Reviewer #1 (Remarks to the Author):

The authors took serious efforts to address my comments. I think this has improved the quality of the paper. In my opinion the manuscript can now be considered for publication in Nature Communications.

Reviewer #3 (Remarks to the Author):

Based on the authors' revised manuscript which has increased the experimental rigor of the contributions, as well as the more complete insights gained from DFT I would support this manuscript's publication. It provides valuable insight on how intermetallic interfaces can significantly alter the reactivity of similar materials.